# Notch1 regulates the initiation of metastasis and self-renewal of Group 3 medulloblastoma

Suzana A. Kahn[1,2,3,4], Xin Wang [5], Ryan T. Nitta[3], Sharareh Gholamin[1,2,3], Johanna Theruvath[1], Gregor Hutter [1], Tej D. Azad[1], Lina Wadi[6], Sara Bolin[1,7], Vijay Ramaswamy[5], Rogelio Esparza[1,3], Kun-Wei Liu[8], Michael Edwards [3,4], Fredrik J. Swartling[7], Debashis Sahoo[9], Gordon Li[3], Robert J. Wechsler-Reya[8], Jüri Reimand[6,10], Yoon-Jae Cho[4], Michael D. Taylor[5], Irving L. Weissman [2,4], Siddhartha S. Mitra [1,2,3,4,11] & Samuel H. Cheshier[1,2,3,4,12]

Medulloblastoma is the most common malignant brain tumor of childhood. Group 3 medulloblastoma, the most aggressive molecular subtype, frequently disseminates through the leptomeningeal cerebral spinal fluid (CSF) spaces in the brain and spinal cord. The mechanism of dissemination through the CSF remains poorly understood, and the molecular pathways involved in medulloblastoma metastasis and self-renewal are largely unknown. Here we show that NOTCH1 signaling pathway regulates both the initiation of metastasis and the self-renewal of medulloblastoma. We identify a mechanism in which NOTCH1 activates BMI1 through the activation of TWIST1. NOTCH1 expression and activity are directly related to medulloblastoma metastasis and decreased survival rate of tumor-bearing mice. Finally, medulloblastoma-bearing mice intrathecally treated with anti-NRR1, a NOTCH1 blocking antibody, present lower frequency of spinal metastasis and higher survival rate. These findings identify NOTCH1 as a pivotal driver of Group 3 medulloblastoma metastasis and self-renewal, supporting the development of therapies targeting this pathway.

[1] Division of Pediatric Neurosurgery, Lucile Packard Children's Hospital, Stanford University School of Medicine, Stanford 94305 California, USA. [2] Institute for Stem Cell Biology and Regenerative Medicine and the Ludwig Cancer Center, Stanford University School of Medicine, Stanford 94305 California, USA. [3] Department of Neurosurgery, Stanford University School of Medicine, Stanford 94305 California, USA. [4] Ludwig Institute for Cancer Research, Stanford University School of Medicine, Stanford 94305 California, USA. [5] Division of Neurosurgery, Arthur and Sonia Labatt Brain Tumor Research Centre, Hospital for Sick Children, University of Toronto, Toronto M5G 0A4 Ontario, Canada. [6] Computational Biology Program, Ontario Institute for Cancer Research, Toronto M5G 0A3 Ontario, Canada. [7] Department of Immunology, Genetics and Pathology, Science for Life Laboratory, Rudbeck Laboratory, Uppsala University, Uppsala 75185, Sweden. [8] Tumor Initiation and Maintenance Program, Sanford Burnham Prebys Medical Discovery Institute, 2880 Torrey Pines Scenic Drive, La Jolla, California 92037, USA. [9] Department of Pediatrics and Department of Computer Science and Engineering, University of California San Diego, San Diego 92093 California, USA. [10] Department of Medical Biophysics, University of Toronto, Toronto M5G 1L7 Ontario, Canada. [11] Department of Pediatrics, Children's Hospital Colorado, University of Colorado, School of Medicine, Room No. P18-4114, Research Complex 1—North MS-8302, 12800 East 19th Avenue, Aurora, Colorado 80045, USA. [12] Division of Pediatric Neurosurgery, Department of Neurosurgery, Primary Children's Hospital and Huntsman Cancer Institute, University of Utah, 100 North Mario Capecchi Drive Suite 3850, Salt Lake City, Utah 84113, USA. These authors jointly supervised this work: Suzana A. Kahn, Siddhartha S. Mitra & Samuel H. Cheshier. Correspondence and requests for materials should be addressed to S.A.K. (email: suzanakahn@gmail.com) or to S.H.C. (email: samuel.cheshier@hsc.utah.edu)

The presence and extent of metastasis are inversely related to the progression-free and overall survival of medulloblastoma patients[1–3]. The mechanism of dissemination through the cerebral spinal fluid (CSF) remains poorly understood and the molecular pathways involved in medulloblastoma metastasis and self-renewal are largely unknown. Cells composing the leptomeningeal metastases and the matched primary medulloblastoma arise from a common transformed progenitor[4]. However, the molecular pathways governing the self-renewal of primary cells and metastatic dissemination are not fully characterized.

The NOTCH1-mediated signaling pathway is critical for mammalian CNS development and plays a crucial role in neural stem cell maintenance and inhibition of neuronal commitment influencing both cell fate decision as well as terminal differentiation of cerebellar granule neuron precursors (GNPs). Recently, significant overrepresentation of NOTCH pathway genes were observed by pathway analysis of recurrent genetic events in Group 3 medulloblastoma[5].

Here we show that loss of TWIST1 resulted in reduced BMI1 expression and inhibition of Group 3 medulloblastoma metastasis. Spinal metastases display increased expression of NOTCH1 and increased levels of NOTCH1 intracellular domain (NICD1), the active form of NOTCH1. Upon orthotopic transplantation, NOTCH1+ cells robustly generate cerebellar tumors and give rise to spontaneous spinal metastases upon primary and secondary transplantation, in contrast to NOTCH1− medulloblastoma cells, which are unable to produce metastases in the primary transplant and do not generate cerebellar tumors in the secondary re-transplant experiments. The NOTCH1 pathway represents a promising target for therapy of Group 3 medulloblastoma.

## Results

### NOTCH1 expression in Group 3 medulloblastoma. Since NOTCH1 activity can drive cancer metastasis by modulating the epithelial–mesenchymal transition (EMT), tumor angiogenesis processes and the anoikis resistance of tumor cells, we asked if NOTCH1 activity regulates Group 3 medulloblastoma metastasis. To explore the metastatic properties of Group 3 medulloblastoma cells, we used two patient-derived Group 3 medulloblastoma primary xenograft lines (MB002[6,7] and Med2112FH, see Methods section), two human Group 3 medulloblastoma cell lines (D283 and D425), and a MYC-driven mouse medulloblastoma cell line (MP[8]) that gave rise to spontaneous leptomeningeal metastasis in vivo, recapitulating the human disease (Fig. 1a, b). These cells were engineered for constitutive expression of GFP and luciferase and orthotopically injected into the cerebella of immune-compromised NOD.Cg-Prkdc[scid] Il2rg[tm1Wjl]/SzJ (NSG) mice (Fig. 1a, b). After confirmation of tumor growth and metastasis by bioluminescent imaging (Fig. 1b), we determined the relative expression of surface NOTCH1 as well as the transcriptionally active NOTCH1 intracellular domain (NICD1) between tumor tissues isolated from primary cerebellar site and metastatic spinal tumor site. We found higher frequencies of cells expressing surface NOTCH1 in tumor cells isolated from spinal metastases compared with tumor cells from the primary tumor sites in all xenografts analyzed (Fig. 1c, d and Supplementary Fig. 1). Furthermore, NICD1 expression, the active form of NOTCH1, was fivefold to tenfold higher in tumor cells from spinal metastases compared with cells from primary tumor sites (Fig. 1e, f). These results were observed in the four human Group 3 medulloblastoma lines and MYC-driven mouse medulloblastoma cell line (Fig. 1c–f and Supplementary Fig. 1). Moreover, Group 3 medulloblastoma spinal metastases expressed higher levels of

NOTCH1 pathway-associated genes (Supplementary Fig. 2a–c and Supplementary Table 1) compared with the primary tumors. Furthermore, in a spontaneous MYCN-driven transgenic medulloblastoma mouse model (GTML)[9], the primary tumor was negative for activated Notch1 and Hes1, whereas both markers were present in the spinal metastasis (Supplementary Fig. 2d). We further found that spinal metastasis samples expressed higher levels of NOTCH1 (Supplementary Fig. 3a) compared with the primary tumors from the same patients. Finally, we analyzed NOTCH1 expression in primary Group 3 medulloblastoma biopsy samples from five patients. Indeed, NOTCH1 is weakly expressed in human primary tumors in the cerebellum (Supplementary Fig. 3b). Our data suggest that Group 3 medulloblastoma cells that metastasize to the spinal cord arise from a distinct subpopulation of tumor cells with increased NOTCH1 pathway activity and expression.

We then analyzed gene expression in a cohort of 46 human primary Group 3 medulloblastoma samples from the Medulloblastoma Advanced Genomics International Consortium (MAGIC) study[10]. To investigate the role of NOTCH1 signaling in downstream gene expression, we divided the samples into two groups (NOTCH1-high and NOTCH1-low) based on median dichotomization of NOTCH1 expression and performed differential expression analysis between the two groups. To further understand the effects of differential NOTCH1 expression in Group 3 medulloblastoma, we conducted a pathway enrichment analysis of the 154 significant genes (q < 0.1) and identified 135 significantly enriched pathways (q < 0.05) (Supplementary Fig. 3c). The major functional themes representing NOTCH1-associated genes included cell migration (n = 24, q = 0.0015), cell motility (n = 19, q = 0.0018), chemotaxis (n = 16, q = 8.15 × 10−4), and regulation of cell adhesion (n = 11, q = 0.0068) (Supplementary Fig. 3c, see Methods section). We also compared NOTCH1 expression in Group 3 medulloblastoma patients, stratifying by presence (M+) or absence (M0) of metastasis at diagnosis (Cavalli Dataset[11]), and found that NOTCH1 expression is higher in M + patients (Supplementary Fig. 3d). Finally, patients with higher NOTCH1 and HES1 (a NOTCH1 target gene) expression in their tumors trended toward decreased overall survival (Supplementary Fig. 4). These data suggest that NOTCH1 potentially plays an important role in human medulloblastoma spinal metastasis.

### NOTCH1 + cells promote Group 3 medulloblastoma metastasis. Due to the increased expression of surface NOTCH1 and NICD1 in Group 3 medulloblastoma spinal metastasis (Fig. 1c–f and Supplementary Fig. 1), we hypothesized that NOTCH1 expressing cells drive spinal metastasis in Group 3 medulloblastoma. As an in vitro surrogate of metastasis, we first compared the invasive properties between Group 3 medulloblastoma cells isolated from primary tumors (predominantly NOTCH1−) and spinal metastasis (predominantly NOTCH1+). In vitro invasion assays revealed that metastatic Group 3 medulloblastoma cells from the spine are more invasive in matrigel chambers than cells from the primary tumors isolated from the cerebellum (Supplementary Fig. 5a). Accordingly, the protein levels of SNAI1, VIM, TWIST1, and CXCR4, known to be involved in cell migration[12], were more highly expressed in spinal metastatic cells than in the primary cerebellar tumors (Supplementary Fig. 5b, c). Finally, NOTCH1+ Group 3 medulloblastoma cells are more invasive in matrigel chambers than NOTCH1−cells (Supplementary Fig. 5d).

To determine whether NOTCH1+ cells exhibited increased metastatic potential in vivo, we sorted the NOTCH1+ and NOTCH1− human medulloblastoma cells from primary cerebellar

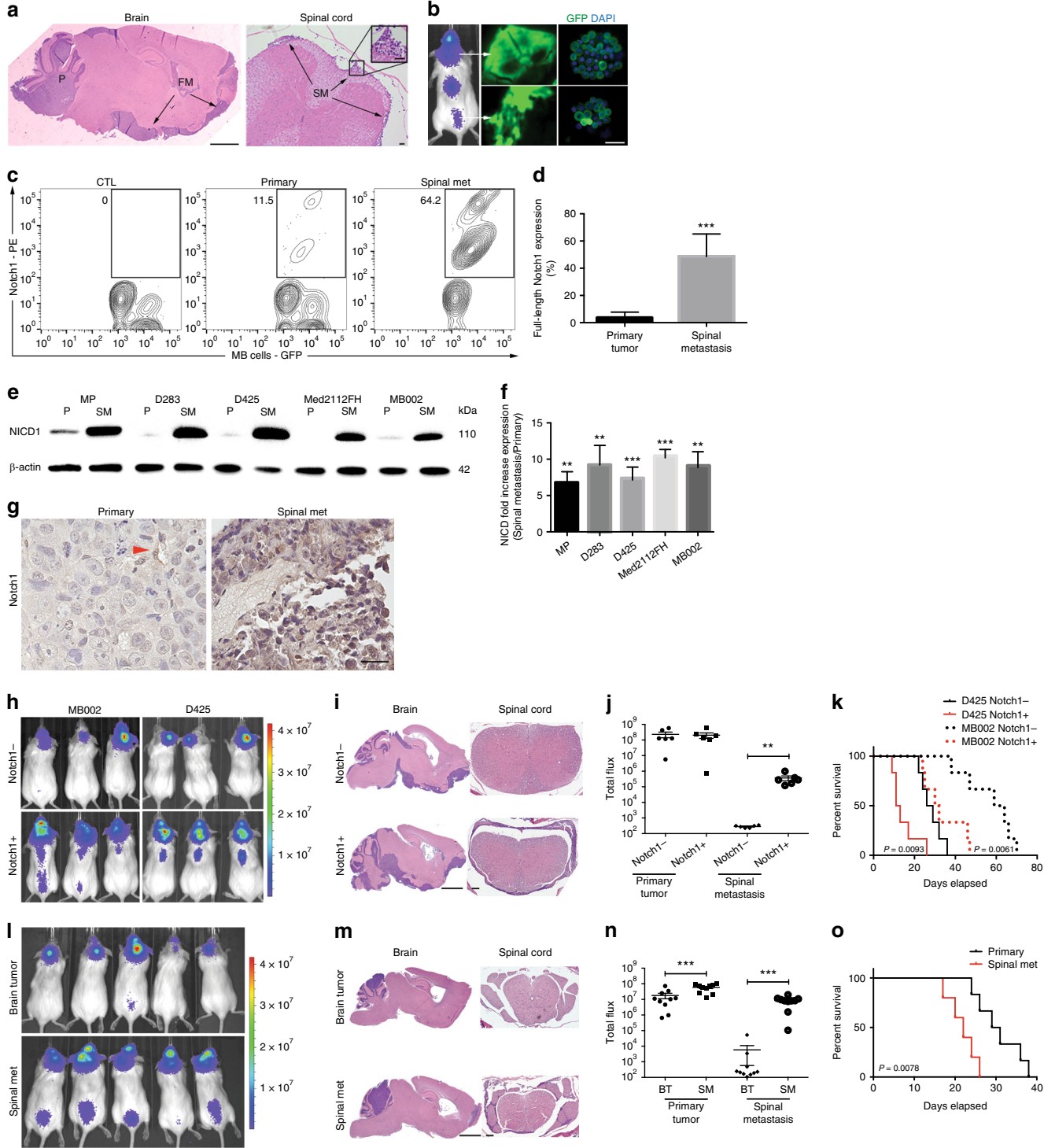

xenografts (Fig. 1b) and separately transplanted them into mouse cerebella (Fig. 1h–j). Although NOTCH1+ and NOTCH1− cells generated similar sizes of primary cerebellar tumors, NOTCH1+ cells produced robust spinal metastases, whereas NOTCH1− cells were unable to produce detectable spinal metastases (Fig. 1h–j). Furthermore, mice transplanted with NOTCH1+ medulloblastoma cells exhibited higher mortality rate than mice transplanted with NOTCH1− cells (Fig. 1k). We then isolated whole medulloblastoma cells from primary mouse tumors in the cerebella (low frequency of NOTCH1+ cells) and from spinal

metastases (high frequency of NOTCH1+cells) and re-transplanted into mouse cerebella. Group 3 medulloblastoma cells isolated from the spinal metastasis generated higher frequency and size of spinal metastases (Fig. 1l–n) and lower survival rate (Fig. 1o) than tumor cells isolated from primary cerebellar tumors. Thus, NOTCH1+ Group 3 medulloblastoma cells exhibit increased invasive and metastatic properties than NOTCH1− cells, supporting the hypothesis that only a pre-existing minor NOTCH1+ subclone of the primary tumor has the ability to metastasize[4].

**Fig. 1** NOTCH1[+] medulloblastoma cells promote Group 3 medulloblastoma metastasis. **a** Representative hematoxylin/eosin (HE) staining of Group 3 medulloblastoma-bearing mouse brain (sagittal section, scale bar, 2 mm) and spinal cord (transversal section, scale bars, 100 μm). P primary tumor, FM forebrain metastasis, SM spinal metastasis. **b** Scheme of in vivo dissemination followed by isolation of primary and metastatic GFP-luciferase-positive Group 3 medulloblastoma cells. **c** Flow cytometry analysis of surface NOTCH1 expression in primary and metastatic Group 3 medulloblastoma cells (MB002) and (**d**) Quantification of five independent experiments. CTL unstained cells. ***$P < 0.001$, Mann–Whitney $U$ test. **e** Immunoblotting for NOTCH1 intracellular domain (NICD1) in cells isolated from primary tumors (P) and spinal metastasis (SM) xenografts generated by two patient-derived Group 3 medulloblastoma cells (MB002 and Med2112FH), two Group 3 medulloblastoma cell lines (D425 and D283), and a MYC-driven mouse medulloblastoma cell line (MP), as well as (**f**) quantification of three independent experiments. **$P < 0.01$, ***$P < 0.001$, Mann–Whitney $U$ test. **g** Representative immunohistochemistry for NOTCH1 in MB002 from primary tumor and spinal metastasis after spontaneous metastasis in mice. The red arrowhead depicts a NOTCH1[+] cell in the primary tumor. Scale bar, 20 μm. **h** Bioluminescence imaging of mice injected with luciferase-expressing NOTCH1[-] or NOTCH1[+] medulloblastoma cells sorted from primary tumors generated by MB002 and D425, HE staining (**i**) and quantification of total flux (**j**) from primary tumors and spinal metastasis in mice injected with NOTCH1[-] or NOTCH1[+] medulloblastoma cells. **$P < 0.01$, Mann–Whitney $U$ test. **k** Kaplan–Meyer survival curves of mice injected with NOTCH1[−] and NOTCH1[+] medulloblastoma cells in MB002 and D425 models. $P$ values are from log-rank test. Scale bars, 100 μm. **l** Bioluminescence imaging. MB002 cells from brain tumors and spinal metastases were isolated from mice and re-transplanted into mouse cerebella. HE staining (**m**) and quantification of total flux (**n**) from primary cerebellar tumors and spinal metastases in mice injected with MB002 cells isolated from brain tumors (BT) or spinal metastases (SM). ***$P < 0.001$, Mann–Whitney $U$ test. **o** Kaplan–Meyer curves of mice injected with Group 3 medulloblastoma cells isolated from primary tumors or spinal metastasis. $P$ value is from log-rank test. Error bars, SD

**NOTCH1–TWIST1–BMI1 axis regulates Group 3 medulloblastoma metastasis.** To test if NOTCH1 was necessary for Group 3 medulloblastoma metastasis, we inhibited its expression with shRNA against NOTCH1 and assessed the ability of medulloblastoma cells to generate spinal metastasis. We transfected Group 3 medulloblastoma cells (D425 and D283) with two constructs: 1) EF1.GFP.T2A.Luc, for constitutive expression of GFP and luciferase and 2) doxycycline/RFP-inducible short hairpin RNA (shRNA) to NOTCH1, for reducing NOTCH1 expression (Supplementary Fig. 6 and Supplementary Table 2). Mice injected with doxycycline-treated, and hence decreased NOTCH1 expressing, medulloblastoma cells (shNotch1 + Dox, Fig. 2a–c and Supplementary Fig. 7a, b) presented lower frequency of spinal metastases compared with control cells (shNotch1−Dox, Fig. 2a-c and Supplementary Fig. 7a, b). Furthermore, mice injected with NOTCH1 silenced Group 3 medulloblastoma cells presented higher survival rates (Fig. 2d and Supplementary Fig. 7c).

To test if NOTCH1 activity was sufficient for Group 3 medulloblastoma metastasis, we restored the levels of NICD1 expression in Group 3 medulloblastoma cells, where *NOTCH1* expression was reduced by shRNA (Supplementary Fig. 6a) by transfecting these cells with three constructs: (1) EF.hICN1.CMV. GFP, for constitutive expression of NICD1 and GFP, (2) doxycycline/RFP-inducible shRNA to NOTCH1, for NOTCH1 activity inhibition (Supplementary Fig. 6), and (3) luciferase, for constitutive expression of luciferase (shNotch1 + Dox + NICD1). Medulloblastoma cells were sorted and orthotopically injected into mouse cerebella. Mice injected with medulloblastoma cells overexpressing NICD1 (shNotch1 + Dox + NICD1) recovered the frequency of spinal metastases (Fig. 2a–c and Supplementary Fig. 7a, b) and reduced the survival rate (Fig. 2d and Supplementary Fig. 7c). Furthermore, in vitro invasion assays revealed that medulloblastoma cells overexpressing NICD1 are more invasive in Matrigel chambers (Supplementary Fig. 8a–c) and express higher levels of SNAI1 and VIM, which are known to be expressed in metastatic cells (Supplementary Fig. 8d).

NICD1 generation is dependent on gamma-secretase (GS) intramembrane activity. Although GS inhibitors (GSIs) have progressed into the clinic[13], GSIs fail to distinguish individual Notch receptors and inhibit other signaling pathways[14]. The dual NOTCH1 and NOTCH2 inhibition leads to severe intestinal toxicity[15], which dramatically limits GSI dosing in oncology patients and reduces effectiveness of GSIs as a therapeutic option. In contrast, anti-NOTCH1 Negative Regulatory Region antibody (anti-NRR1) is specific to NOTCH1 and inhibits only the

NOTCH1 signaling pathway through stabilizing NRR1 quiescence[16,17]. In essence, it potently inhibits its cognate paralogue but not other NOTCH receptors. This NOTCH1-specific inhibition avoids the intestinal toxicity observed when both NOTCH pathways are inhibited and suggests a clear advantage of anti-NRR1 over pan-NOTCH inhibitors[16,17]. As intrathecal methotrexate is the standard of care for infants in Europe[18] and intraventricular administration maximizes the amount of therapeutic material in the cerebrospinal fluid, we administered anti-NRR1 intrathecally in Group 3 medulloblastoma-bearing mice (Supplementary Table 3). Anti-NRR1-treated mice presented lower frequency of spinal metastases (Fig. 2e–g and Supplementary Fig. 7d, e) and higher survival rate (Fig. 2h and Supplementary Fig. 7f). Although the effect of anti-NRR1 is less evident when mice are treated after spinal metastases are formed, spinal metastases-bearing mice intrathecally treated with anti-NRR1 presented lower frequency of spinal metastases and higher survival rate. (Supplementary Fig. 7g, h). Interestingly, NICD1 overexpression, but not NICD2, recovers the invasive ability of anti-NRR1-treated medulloblastoma cells (Supplementary Fig. 9a). This finding, together with the fact that NOTCH2-silenced, but not NOTCH1-silenced, medulloblastoma cells show increased late apoptosis (Supplementary Fig. 10) support initial findings that NOTCH1 and NOTCH2 have different roles on embryonal brain tumors[19]. These data indicate that the activation of the NOTCH1 signaling pathway increases Group 3 medulloblastoma spinal metastasis and point to the inhibition of this pathway as a potential therapeutic target.

BMI1, a component of the Polycomb Repressive Complex 1 (PRC1)[20], is implicated in the pathogenesis of medulloblastoma[21] and has been associated with poor outcome in Group 3 medulloblastoma[21]. Moreover, prior work has identified a BMI1-driven gene signature that predicts metastasis, tumor progression, and death from cancer across 11 cancer subtypes[22]. Since (1) NOTCH intracellular domain directly binds to the *TWIST1* promoter[23,24], (2) *BMI1* is a direct transcriptional target of TWIST1[25], and (3) BMI1 has been shown as a NOTCH1[26] and TWIST1[25] downstream target, we hypothesized that the TWIST1–BMI1 axis[24,27] is involved in NOTCH1-induced Group 3 medulloblastoma metastasis. We found that NICD1 overexpression increases TWIST1 and BMI1 expression (Fig. 2i, j). On the other hand, both TWIST1- and BMI1-silenced medulloblastoma cells present similar levels of NICD1 expression as compared with control (Fig. 2i, j), suggesting that NOTCH1 is upstream of TWIST1 and BMI1. TWIST1-silenced medulloblastoma cells present reduced BMI1 expression, while BMI1-silenced

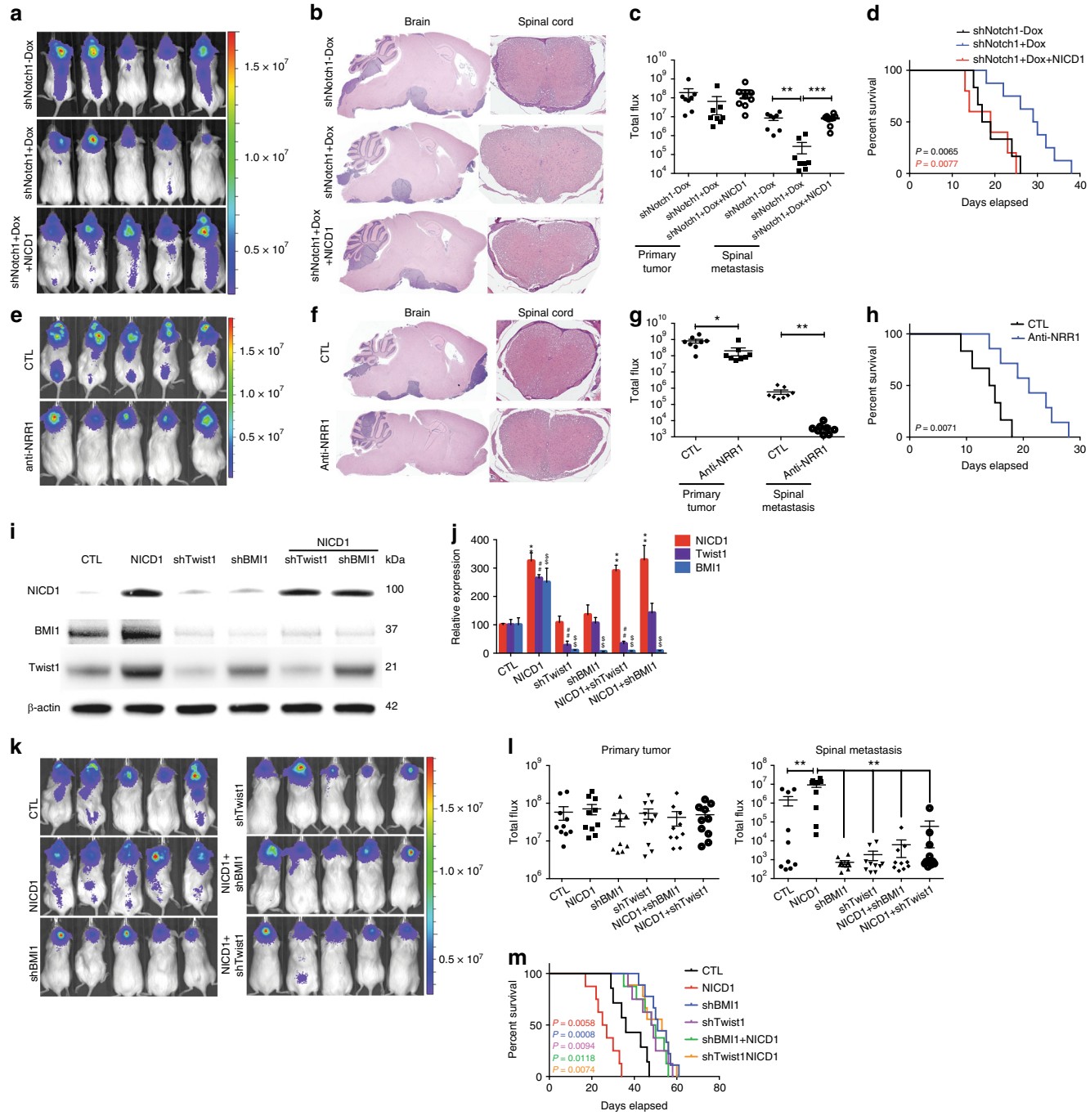

medulloblastoma cells present similar levels of TWIST1 expression as compared with control (Fig. 2i, j). Finally, TWIST1-silenced and BMI1-silenced cells have significantly decreased metastatic propensity in vivo (Fig. 2k–m), regardless of NICD1 overexpression (Fig. 2k–m). Together, these findings indicate that the NOTCH1-induced TWIST1 activation contributes to the activation of BMI1, therefore inducing Group 3 medulloblastoma metastasis. Whether this pathway has intermediate players, or NICD1 directly binds to TWIST1 promoter and TWIST1 to BMI1 promoter is still unknown and would be of great interest for future studies. We further investigated the NOTCH1–TWIST1–BMI1 axis in regulating MYC expression. NOTCH1-silenced medulloblastoma cells express lower levels of MYC (Supplementary Fig. 2e). Interestingly, TWIST1-silenced

medulloblastoma cells present similar levels of MYC expression as compared with control, suggesting that NOTCH1, but not TWIST1, is upstream of MYC. This finding indicates that NOTCH1–MYC and NOTCH1–TWIST1–BMI1 act at different axes in Group 3 medulloblastoma metastasis.

**NOTCH1 promotes Group 3 medulloblastoma self-renewal.** Group 3 medulloblastoma relapse and metastasis remain major obstacles for improving overall patients survival, which may be due at least in part to the existence of medulloblastoma-initiating cells[28] that use many of the same signaling pathways found in normal stem cells, such as Wnt, Hedgehog, and Notch[28,29]. Indeed, the major functional themes representing NOTCH1-

**Fig. 2** NOTCH1 signaling regulates Group 3 medulloblastoma metastasis. **a–d** shRNAs specific to NOTCH1 (shNotch1) were introduced through lentiviral infections into luciferase-expressing D425 cells. 2 μg mL$^{-1}$ of Dox was used to induce the expression of shRNA (shNotch1 + Dox). The un-induced infected cells (shNotch1-Dox) were used as controls. Recovery of NOTCH1 expression was performed by overexpressing NICD1 in NOTCH1-silenced Group 3 medulloblastoma cells (shNotch1 + Dox + NICD1). Bioluminescence imaging (**a**), hematoxylin/eosin staining (**b**), and quantification of total flux (**c**) from primary tumors and spinal metastases in mice injected with infected cells. ** $P < 0.01$, Mann-Whitney $U$ test. **d** Kaplan–Meyer survival curves of mice injected with Group 3 medulloblastoma cells down-expressing NOTCH1 (shNotch1 + Dox, blue), control (shNotch1-Dox, black), or with recovered NICD1 expression (shNotch1 + Dox + NICD1, red). $P$-value is from log-rank test. **e** Bioluminescence imaging of mice injected with luciferase-expressing D425 cells and intrathecally treated with anti-NRR1 or vehicle (CTL). Hematoxylin/eosin staining (**f**) and quantification of total flux (**g**) from primary tumors and spinal metastases. *$P < 0.05$, **$P < 0.01$, Mann-Whitney $U$ test. **h** Kaplan–Meyer survival analysis of mice intrathecally treated with anti-NRR1 or vehicle (CTL). $P$ value is from log-rank test. **i** Immunoblotting for NICD1, TWIST1 and BMI1 in D425 cells control (CTL); overexpressing NICD1 (NICD1); down-expressing Twist1 (shTwist1); down-expressing BMI1 (shBMI1), overexpressing NICD1 and down-expressing TWIST1 (NICD1 + shTwist1); and over-expressing NICD1 down-expressing BMI1 (NICD1 + shBMI1). kDa, Kilodaltons. β-actin was used as loading control. **j** Quantification of NICD1 (red), TWIST1 (purple) and BMI1 (blue) expression relative to β-actin from three independent experiments. **$P < 0.01$ versus NICD1 control; ##$P < 0.01$ versus Twist1 control; $$$P < 0.01$ versus BMI1 control, Mann-Whitney $U$ test. Bioluminescence imaging (**k**), quantification of flux values (**l**), and survival curves (**m**) from mice injected with luciferase-expressing D425 cells over-expressing NICD1 (NICD1, red); control (CTL, black); down-expressing TWIST1 (shTwist1, purple); down-expressing BMI1 (shBMI1, blue); over-expressing NICD1 and down-expressing Twist1 (NICD1 + shTwist1, orange); over-expressing NICD1 and down-expressing BMI1 (NICD1 + shBMI1, green). Error bars, SD

associated genes in our analysis included neurogenesis ($n = 24$, $q = 1.78 \times 10^{-4}$), cell morphogenesis ($n = 20$, $q = 0.0044$), multicellular organism development ($n = 22$, $q = 2.77 \times 10^{-4}$), and embryo development ($n = 8$, $q = 0.038$) (Supplementary Fig. 3c, see Methods section). To further assess the role of NOTCH1 in Group 3 medulloblastoma stem cell properties, we sorted the NOTCH1$^+$ and NOTCH1$^-$ medulloblastoma cells from primary tumors of the cerebella and metastatic tumors of the spine of MB002 and D425 xenografted mice. NOTCH1$^+$ medulloblastoma cells from both primary and metastatic tumors exhibited higher secondary neurosphere forming ability (Fig. 3a and Supplementary Fig. 11a) and increased clonogenicity (Fig. 3b and Supplementary Fig. 11b) than NOTCH1$^-$ cells, as determined by limiting dilution analysis. Also, NOTCH1$^+$ medulloblastoma cells from primary tumors express higher levels of CD15 (Supplementary Fig. 11c) and Ki67 (Supplementary Fig. 11d) than NOTCH1$^-$ medulloblastoma cells from the same tumors. Accordingly, Group 3 medulloblastoma cells treated with anti-NRR1 exhibited reduced secondary neurosphere forming ability, clonogenicity, and migration (Supplementary Fig. 9). Of note, similarly to NOTCH1 (Fig. 1g and Supplementary Fig. 3a, b), BMI1 and Nestin are weakly expressed in Group 3 medulloblastoma primary tumors from D425 xenografted mice (Supplementary Fig. 12).

To determine whether NOTCH1$^+$ cells represent medulloblastoma self-renewing tumor stem cells, we performed in vivo serial transplant assays. Human Group 3 medulloblastoma cerebellar xenografts were harvested and separated into NOTCH1$^+$ and NOTCH1$^-$ cells by flow cytometry. These separated tumor cells were then injected into mouse cerebella (first passage in vivo) (Figs. 1h and 3c). After establishment of the tumors generated by NOTCH1$^+$ and NOTCH1$^-$ cells, the primary recipient cerebellar tumors were harvested and the NOTCH1$^+$ and NOTCH1$^-$ cells were resorted from these primary tumors and re-transplanted into the cerebella of secondary recipients (second passage in vivo) (Fig. 3c–f). Secondary transplants of NOTCH1$^-$ cells were unable to generate new primary tumors when reinjected into mouse cerebella, contrasting with the NOTCH1$^+$ cells which robustly formed brain tumors and spinal metastases upon reinjection (Fig. 3c–f). These findings indicate that NOTCH1 is not only involved in the generation of spinal metastasis, but also in the self-renewal/maintenance of the primary tumor. Of note, tumors generated by NOTCH1$^-$ cells were unable to produce NOTCH1$^+$ cells, whereas NOTCH1$^+$ cells were able to produce NOTCH1$^+$ and

NOTCH1$^-$ cells (Supplementary Fig. 13), indicating a cell lineage of self-renewing NOTCH1$^+$ cells capable of giving rise to NOTCH1$^+$ and NOTCH$^-$ cells, whereas NOTCH$^-$ cells can only give rise to other NOTCH$^-$ cells in an unsustained manner. Accordingly, mice injected with the second in vivo passage of NOTCH1$^-$ cells presented higher survival rate (Fig. 3f). Finally, to assess the tumorigenic potential of more limited cell number injection, human Group 3 medulloblastoma cerebellar xenografts were harvested, and 100 cells of either NOTCH1$^+$ or NOTCH1$^-$ were injected into mouse cerebella (Fig. 3g–i). Tumor formation was observed in nine of the ten mice injected with NOTCH1$^+$ medulloblastoma cells, as opposed to tumor formation in only one of the 10 mice injected with NOTCH1$^-$ higher survival rate medulloblastoma cells (Fig. 3g–i). Metastasis was only present in the NOTCH1$^+$injected setting. Whether NOTCH1$^+$ primary and NOTCH1$^+$ metastatic medulloblastoma cells represent the same population of cells or contain genomic differences is still unknown. These data suggest that NOTCH1$^+$ medulloblastoma cells are more capable of initiating spinal metastasis but are also responsible for Group 3 medulloblastoma self-renewal.

We have unified metastatic activity with self-renewal activity in Group 3 medulloblastoma under one common molecular pathway–NOTCH1. Our data from in vivo spontaneous metastasis across five Group 3 human medulloblastoma lines, a MYC-driven mouse medulloblastoma cell line, and from human medulloblastoma gene expression database analysis, show that medulloblastoma spinal metastases present higher NOTCH1 pathway expression than matched primary tumors and that NOTCH1 pathway activation increases medulloblastoma metastasis and reduces survival. Furthermore, inhibition of NOTCH1 pathway activity reduces medulloblastoma metastasis and increases survival. We also show that NOTCH1$^+$ medulloblastoma cells are responsible for the self-renewal of both the primary and metastatic tumor compartments of Group 3 medulloblastoma. Our findings suggest that NOTCH1-induced medulloblastoma metastasis occurs through TWIST1-induced BMI1 activation (Fig. 3j). In conjunction with the low adverse effects of specifically inhibiting NOTCH1[15,17,116] this pathway represents a promising target for therapy of Group 3 medulloblastoma.

## Methods
**Primary tissue dissociation and generation of primary cell lines**. Group 3 medulloblastoma tissue samples were obtained under IRB 18672 after patient consent at the Lucile Packard Children's Hospital (Stanford, CA) in accordance with institutional review board protocols. Pathologies of tumors were confirmed

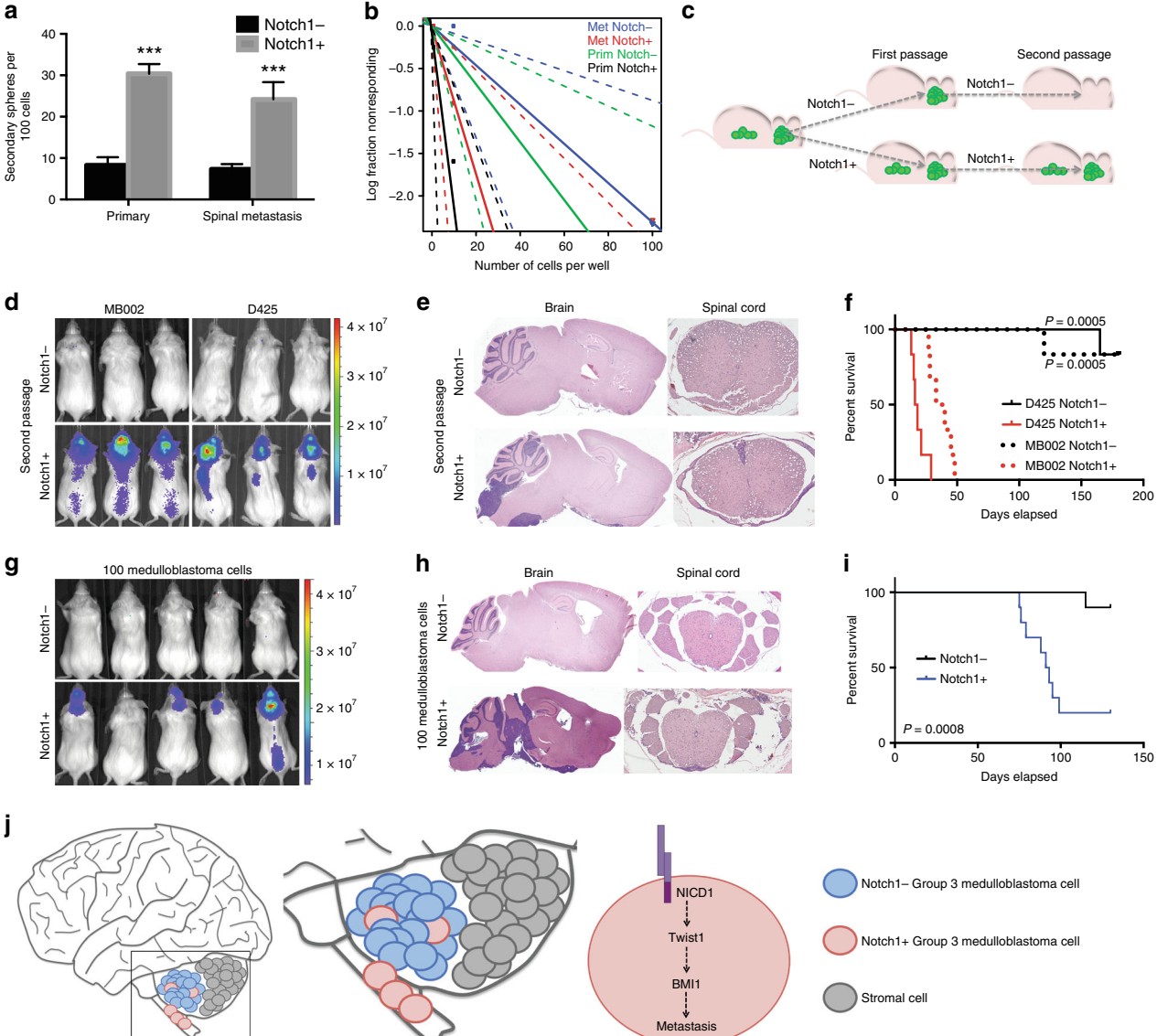

**Fig. 3** NOTCH1+ cells promote Group 3 medulloblastoma self-renewal. **a** In vitro extreme limiting dilution assays comparing the frequency of tumorsphere formation between NOTCH1− and NOTCH1+ Group 3 medulloblastoma cells isolated from primary tumors and spinal metastases. ***$P < 0.001$, Mann–Whitney $U$ test. **b** Quantitative analysis of the frequency of colonies formed by NOTCH1− and NOTCH1+ Group 3 medulloblastoma cells isolated from primary tumors and spinal metastases. ELDA software (http://bioinf.wehi.edu.au/software/elda/) was used to calculate medulloblastoma stem cell frequency and assess statistical significance. Prim NOTCH1- (green) versus Prim NOTCH1+ (black) ($P = 0.02$) and Met NOTCH1− (blue) versus Met NOTCH1+ (red) ($P = 0.04$). **c** Scheme of NOTCH1− and NOTCH1+ in vivo sorting and passages. **d** Bioluminescence imaging and hematoxylin/eosin staining of brains and spines (**e**) from mice injected with the second in vivo passage of NOTCH1− or NOTCH1+ sorted from primary tumors generated by MB002 and D425. **f** Kaplan–Meyer survival curves of mice injected with the second in vivo passage of NOTCH1− and NOTCH1+ in two models (MB002 and D425). *P*-values are from log-rank test. Bioluminescent imaging (**g**), hematoxylin/eosin staining of brains and spines (**h**), and survival curves (**i**) from mice injected with 100 NOTCH1− or NOTCH1+ medulloblastoma cells (D425). **j** Proposed model: NOTCH1-induced medulloblastoma metastasis occurs trough the activation of the NICD1-TWIST1-BMI1 axis. Error bars, SD

upon histopathology analysis by institutional, board-certified neuropathologists prior to further analysis. Medulloblastoma samples were enzymatically dissociated to single cells by collagenase IV (1 mg mL$^{-1}$) and DNase I (250 units mL$^{-1}$) and cells were plated in neural stem cell expansion media (NSCEM) consisting of Neurobasal (-A) (Invitrogen), B27(-A) (Invitrogen), human-bFGF (20 ng mL$^{-1}$) (Shenandoah Biotech), human-EGF (20 ng mL$^{-1}$) (Shenandoah Biotech), human recombinant LIF (Millipore) (as required) and Heparin-sulfate (10 ng mL$^{-1}$). Cells were grown for two passages and infected with EF1-GFP-T2A-Luciferase (Systems Biosciences, BLIV503MN-1) and allowed to reform spheres. Cells were then double sorted for GFP and further cultured in NSCEM.

**Medulloblastoma cell cultures**. D283 and D425 were generously provided by Dr. Darrell Bigner (Duke University, Durham, NC). MYC-driven mouse medulloblastoma cell line, MP[8] was generously provided by Dr. Robert Wechsler-Reya.

MB002 cells[6] were derived postmortem from the leptomeningeal compartment of a child with metastatic, treatment-refractory (chemotherapy only) medulloblastoma. Med2112FH was from the Brain Tumor Resource Laboratory (http://www.btrl.org), D425s line is a subclone derived from D425 line. All medulloblastoma lines used presented MYC amplification. MYC amplification in the MB002 cells was confirmed with NanoString nCounter v2 Cancer CN Codeset. The patient-derived medulloblastoma primary cells were authenticated using sequence-tagged site (STS) fingerprinting and all medulloblastoma cell lines were authenticated by short tandem repeat profiling and by PCR-based mycoplasma test. Each line was evaluated for its ability to fully recapitulate the tumor of origin by orthotopic transplantation into mouse cerebella.

**Orthotopic transplantation of medulloblastoma cells**. NOD.Cg-*Prkdc*$^{scid}$ *Il2rg*$^{tm1Wjl}$/SzJ (NOD-SCID gamma) mice were housed in specific pathogen-free

conditions at a barrier facility at the Lokey Stem Cell Building (SIM1) at Stanford School of Medicine (Stanford, CA). All animal handling, surveillance, and experimentation was performed in accordance with and approval from the Stanford University Administrative Panel on Laboratory Animal Care (Protocol number 26548). Early passages spheres of medulloblastoma cells were transduced with a GFP and luciferase encoding lentivirus and double sorted to obtain a pure luciferase-expressing population. Medulloblastoma cells were injected into at coordinates 2 mm posterior to lambda on midline and 2 mm deep into 4–6-week-old NOD-SCID gamma mice. Thirty thousand medulloblastoma cells were injected per mouse, except for Fig. 3g–I where only 100 medulloblastoma cells were injected. Tumor formation was followed by bioluminescence imaging on IVIS spectrum (Caliper Life Science) and quantified with Live Image 4 software. Blinding of the investigator was performed when assessing results. The bioluminescent images were acquired by a second scientist, who was not informed about the identity of the analyzed groups.

**Intrathecal treatment.** Once tumor masses were detected in the brain, mice were randomized in two groups based on flux values, prior to treatment. Anti-NRR1 was purified with PD Columns from GE Heathcare, according to manufactures instructions. To achieve continuous intraventricular CNS administration of anti-NRR1, pumps (Alzet Co., Model 1004, flow rate 0.11 μL hr$^{-1}$) were loaded with 1 μg μL$^{-1}$ of antibody or PBS (CTL). Pumps were coupled to brain infusion kits (Alzet Co., Model 8851) and primed for 48 h at 37 °C, 5% $CO_2$. Osmotic pumps were planted subcutaneously on the dorsum, slightly caudal to the scapulae. Using a stereotaxic apparatus, brain cannulae were inserted intraventricular per pre-defined coordinates (on the coronal suture, 2 mm right lateral to midline, 4 mm into the lateral ventricle) following removal of periosteal connective tissue and secured with dental cement (Stoelting Co.) and continuous sutures in the skin (Ethilon 5-0, Johnson & Johnson). Pumps containing anti-NRR1 stayed implanted until mice were sacrificed (between 10 and 50 days).

**Flow cytometry analysis.** Medulloblastoma tumorspheres were dissociated to single cells and stained with anti-Notch1-PE, anti-Notch2-APC or anti-CXCR4-PE/Cy7 (Miltenyi). Hematopoietic and endothelial cells were gated out using a lineage mixture of Pacific blue-conjugated anti-CD45, anti-CD31. For gating out mouse cells we used anti-H2Kb and anti-H2Kd, anti-Ter119, anti-CD31 and anti-CD45 antibodies. Flow cytometric analysis and cell sorting were performed on a BD FACS aria II (Becton Dickinson). Appropriate isotype and fluorescence minus-one controls were used to define the background gates.

**Immunohistochemistry.** Spines from tumor-bearing mice were decalcified with Cal-Ex™ II Fixative/Decalcifier (Fisher Chemical) for 24 h prior to formalin fixation. Primary and metastatic human medulloblastoma tissues, orthotopic tumor bearing mouse brains and spines were fixed in formalin and embedded into paraffin blocks. Tissue sections (8 μm) were processed for standard hematoxylin and eosin staining or processed for Notch1 staining (Abcam ab52627). Antigen retrieval was done in citrate buffer and blocking was done in 10% horse serum. The secondary antibody (goat anti-rabbit biotinylated) and the streptavidin-HRP were from Jackson ImmunoResearch Labs and DAB chromogen was from Vector Labs. Images were acquired using a Nikon E1000M microscope with Spot Flex camera. Paraffin embedded tissue (brain & spine) from a MYCN-driven transgenic mouse[9], with primary brain tumor and a spinal metastatic compartment, was stained using hematoxylin/eosin for overall histopathology and for activated Notch1 (Abcam) and Hes1 (R&D). Stainings were performed using the ImmPRESS (Vector Laboratories) system and subsequent DAB (Activated Notch1)/VIP (Hes1) enzymatic substrates.

**Lentiviral infections.** The lentivirus pWPT-NICD1 was generated by subcloning the activated Notch1 intracellular domain (EF.hICN1.CMV.GFP Addgene plasmid 17623) into the pWPT GCCACC backbone[31]. Lentiviral infections were carried out using the packaging plasmids, psPAX2 and pMD2.G and were transfected into HEK293 cells[32]. Forty-eight hour after infection, the mCherry$^+$ or GFP$^+$ cells were sorted. Lentiviral shRNA to NOTCH1 and BMI1 were generated using the doxycycline-inducible plasmid pTRIPZ (Thermoscientific) and shRNA for TWIST1 was generated using pGIPZ (Thermoscientific). ShNotch1 was a TRIPZ Inducible Lentiviral shRNA clone V3THS_332663 from Dharmacon. Lentiviral shRNA to Notch2 was from Dharmacon. Anti-NRR1 plasmid[17] was from Addgene (E6-pBIOCAM5). Lentiviral infections were carried out as described above. Thirty-six hour after infection, the cells were selected using 0.5 μg mL$^{-1}$ of puromyocin for 3 days. The shRNA expression was induced using 2 μg mL$^{-1}$ of doxycline for 1–10 days. The knockdown of NOTCH1, TWIST1, and BMI1 expression was verified by Western blot analysis.

**Cell proliferation assays.** Cells were plated in 96-well plates at $2 \times 10^4$ cells/well and treated with the appropriate compounds at 37 °C, 5% $CO_2$. Cell viability was assessed using reduction of WST-1 (4-[3-(4-iodophenyl)−2-(4-nitrophenyl)−2H-5-tetrazolio]−1,3-benzene disulfonate, Roche, France) to water-soluble formazan. At the end of the incubation period, 10% (v/v) WST-1 was added to the culture media, and the cells were further cultured for 3 h. The absorbance was measured at 430 nm in a microplate reader (Expert Plus V1,4 ASYS). Cell viability was also verified with cell counting following addition of Trypan blue (SIGMA) at a final concentration of 0.1% (v/v), and routine examination of the cells under phase contrast microscopy. Data were normalized relatively to the vehicle-treated controls.

**Western blot.** Protein extracts from cells were harvested and immunoblotted[32]. The following antibodies were used for immunoblotting: Notch1 (Val 1744; D3β8), BMI1 (D20B7), Snail (C15D3), Vimentin (D21H3), and E-cadherin (24E10) were from Cell Signaling Technology. Twist1 (3E11) was from Abnova and β-actin (BA3R) was from Thermo Scientific. Enhanced Chemiluminescence Substrate (PerkinElmer) and Gene GNOME (Syngene) was used for visualization. Chemiluminescence signals were quantitated using NIH Image J (National Institutes of Health).

**Real-time qPCR and PCR array.** Total RNA was isolated from medulloblastoma cells using TRIZol reagent (Life Technologies) according to the manufacturer's instructions. cDNA was synthesized and reverse transcribed with the RT$^2$ First Strand Kit (Qiagen, Valencia, CA, USA) according to the manufacturer's instructions. Real-time qPCR was performed using RT$^2$ SYBR® Green ROX ™ qPCR Mastermix (Qiagen). Primer sets for *HES1*: (F) 5′-AGG CGG ACA TTC TGG AAA TG-3′, (R) 5′-CGG TAC TTC CCC AGC ACA CTT-3′ were purchased from Integrated DNA Technologies (IDT). *Hprt1*: (F) 5′-CAC CCT TTC CAA ATC CTC AG-3′ (R) 5′-CTC CGT TAT GGC GAC CC-3′ was used as a housekeeping gene. The $2^{\Delta\Delta Ct}$ method was used to calculate fold changes in gene expression normalized to hypoxanthine phosphoribosyltransferase 1 (*Hprt1*). Transcript levels of Notch-signaling related genes were analyzed as a part of the Human Notch Signaling Targets RT$^2$ Profiler Custom PCR Array (Qiagen). We normalized using the average of five different housekeeping genes, β-actin (*Actb*), glyceraldehyde-3-phosphate dehydrogenase (*Gapdh*), beta-2-microglobulin (*Btm*), HRPT1, and Ribosomal protein, large, P0 (*Rplp0*).

**Notch1 pathway enrichment analysis.** Expression data for 46 subtype Group 3 medulloblastoma samples were obtained from the MAGIC study[10]. Multiple expression values corresponding to a single gene were aggregated using the mean. Samples were divided into two groups of equal size (*NOTCH1*-high and *NOTCH1*-low) by median dichotomization such that the first group had above-median *NOTCH1* expression values and the second group had below-median values. Differential expression analysis between the two *NOTCH1* groups was performed using non-parametric Wilcoxon tests. Significant genes were obtained using an FDR-adjusted *p*-value of 0.10.

Pathway enrichment analysis was performed with g:Profiler[30] and visualized as Enrichment Map (http://baderlab.org/Software/EnrichmentMap/). Pathways with corrected *p*-value *p* < 0.05 were considered significant. Genes were ranked by significance of differential expression (adjusted *p*-value) and analyzed with the ordered query setting of g:Profiler. Biological processes from the Gene Ontology and pathways from Reactome were included in the enrichment analysis and other data sources were excluded. Enriched categories were further filtered: pathways and processes with less than two differentially expressed genes were discarded. Enrichment map was manually revised to annotate the most representative biological themes.

**In vitro cell invasion assay.** 24-well invasion chambers (Corning) were coated with 100 μL of matrigel basement membrane matrix (354234, Corning) for 12 h at 37 °C. The chambers were placed in 24-well plate wells containing NSCEM with 10% fetal bovine serum (FBS) and $5 \times 10^4$ medulloblastoma cells were plated in the inner side of the chambers, in FBS-free media. After 16 h at 37 °C, 5% $CO_2$, the inserts were washed, fixed with 4% paraformaldehyde (PFA), and stained with Giemsa (Sigma-Aldrich). Non-invasive cells, in the inner part of the inserts, were removed with cotton swabs. Images of the invasive cells were acquired using a Nikon E1000M microscope with Spot Flex camera.

**Statistical analysis.** Statistical analysis tests are specified in figure legends. The level of significance was set at *p* < 0.05, and results are shown as mean ± SD of at least three independent experiments performed with at least triplicates per condition. For in vivo experiments, at least eight mice per cohort were used. Samples or animals were not excluded from the analysis. Statistical analyses were carried out with Prism 6.0 software (GraphPad).

## Data availability

The data that support the findings of this study are available from the corresponding authors upon reasonable request.

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

## Acknowledgements

This work was supported by the Pew Latin American Fellowship (S.A.K.), Price Family Charitable Fund, Center for Children's Brain Tumors (S.G, S.S.M, and S.H.C.), American Brain Tumor Foundation (S.H.C.), Matthew Larson Foundation for Pediatric Brain Tumors (S.A.K. and S.H.C.), Virginia and D.K. Ludwig Fund for Cancer Research (I.L.W. and S.H.C.), Lucile Packard Foundation for Children's Health, Child Health Research Institute at Stanford Tashia and John Morgridge Faculty Scholarship in Pediatric Translational Medicine NIH-NCATS-CTSA UL1 TR001085 (S.H.C.). German Cancer Aid (Deutsche Krebshilfe) P-91650709 (J.T.). Ty Louis Campbell Foundation St. Baldrick's Foundation Award (S.HC.) Gifts from Kathryn S.R. Lowrey, George Landegger, Rider and Victoria McDowell, Charles Comey and Judith Huang, and Colin and Jenna Fisher (S.H.C.). Canadian NSERC Discovery Grant RGPIN-2016-06485 (L.W.). Operating Grant 21089 of the Cancer Research Society of Canada (J.R.). R01 NS096368-02 (R.J.W.-R.).

## Author Contributions

I.L.W., M.D.T., Y.J.C., G.L., M.E., S.H.C, S.S.M, and S.A.K designed the experiments. S.G., J.T., G.H, S.B., F.J.S., and S.A.K performed the in vivo experiments. R.T.N., K.W.L, and R.J.W.R generated cell lines (overexpressing/downexpressing NOTCH1 and NOTCH2, and MP mouse medulloblastoma cell line). X.W., T.D.A., L.W., D.S., V.R. and J.R. generated and analyzed the RNAseq data. S.S.M., S.H.C., X.W., R.T.N., T.D.A., R.E., L.W., S.B., V.R. and S.A.K analyzed the experiments. The manuscript was written by S.A.K with input from all authors.

## Additional information

**Competing interests:** The authors declare no competing interests.

