## [Peer Review File · Nature Communications]

Reviewers' comments:

Reviewer #1 (Remarks to the Author):

This manuscript claims that Notch1, but not Notch2, is largely responsible for leptomeningeal spread of Group 3 medulloblastomas. The authors further claim that transcriptional induction of BMI1 through activation of Twist1 is the mechanism by which medulloblastoma cells gain metastatic potential and possibly self-renewal. The study is well done and some specific findings are novel even if the role of NOTCH in medulloblastoma has been debated for over a decade.

Several topics could be better addressed in this manuscript:

The use of models was inconsistent throughout the paper and in some cases problematic. No data is presented or cited to indicate that D283 or D425 cell lines are truly Group 3. Given that a frequently used medulloblastoma line (DAOY) is said to be isolated from a 4 year old boy, yet has two x chromosomes and no Y chromosomes, it is critical to validate these 'workhorse' cell lines if they are used in papers intended to influence clinical trial design.

There is no explanation why some cell lines or models, but not others, were used for certain experiments. If additional relevant data was generated from some of these models but not included in the manuscript, it should be added, even if inconsistent with the current narrative.

The focus of this paper is on Group 3 medulloblastoma, but it is not clear whether there is a link to MYC/MYCN amplification, which is the subset of Group 3 tumors with the poorest prognosis. A link between the NOTCH pathway and MYC was previously made (PMID: 24708907).

Overexpression of ATOH1 caused SHH-driven medulloblastomas to form leptomeningeal disease and become more aggressive (PMID: 28490517). Given that this prior publication could diminish the novelty of the current manuscript, it seems important to either address whether activation of NOTCH1 in non-Group 3 medulloblastomas might lead to the less common leptomeningeal metastases cases in the other groups. While it is too much to ask for the current manuscript, it might be worth assessing NOTCH1 pathway in human leptomeningeal disease compared to primary tumor (autopsy cases) from all groups. An easier step for the current manuscript would be to assess notch pathway signatures from patients who were M+ vs M0 regardless of molecular subclass. The reviewer recognizes that the tissue derived from the primary tumor may not show the full extent of NOTCH pathway activation in mets.

The authors argue that the failure of a pan-NOTCH inhibitor in human clinical trials was due to the lack of therapeutic window when both NOTCH1 and NOTCH2 are inhibited. This claim is refuted by a clinical trial in which NOTCH signal was diminished in 6 of 7 children for whom tissue was available 24 hours after MK-0752 dosing at doses that were well tolerated. Furthermore, bi-allelic knockout of NOTCH1 and NOTCH2 had no effect pathogenesis of Shh-driven medulloblastomas in murine models, suggesting that the pathway may be unimportant in some types of medulloblastoma (PMID: 20440271) and that clinical trials should focus specifically on molecularly selected patients who are most likely to benefit from NOTCH1 inhibition. With this in mind, it is important to add depth to the discussion with regard to the potential use of anti-NRR1 antibodies intrathecally in human patients with leptomeningeal disease. Intrathecal medications are not typically used in this population because CSF flow is impaired by metastases. Antibodies do not cross the BBB. So, what, if anything, are the authors proposing in terms of future clinical trials?

Likewise, for this work to be important in clinical translation, it would be important to know if anti-NRR1 antibodies affect metastases that have already formed. Prevention of metastases could be important, but effective treatment of established metastases would be much more valuable. Additional experiments showing the effect of anti-NRR1 on established metastases would be valuable if technically feasible given the impaired CSF flow and the challenges of quantifying

leptomeningeal disease burden before and after treatment in living mice. Perhaps inducible knock-out/knock-down would be a feasible method.

The statement that elevated NOTCH1 and HES1 “trended” toward worse survival in patients involved in the MAGIC consortium lacks statistical and biological rigor. Was this observed in other large datasets? Why was HES1 chosen as the only NOTCH1 target gene for this analysis? Given that HES1 is activated by other forms of NOTCH and that there are many other targets of NOTCH1, the selection of HES1 seems somewhat arbitrary. Since the intracellular domain of NOTCH is the active domain, what is the link between gene expression and NICD protein fragment?

In summary, interesting, important and experimentally well supported conclusions with potential to influence how clinical trialists and biologists view medulloblastoma leptomeningeal disease.

Reviewer #2 (Remarks to the Author):

Kahn et al. describe the role of Notch signaling as a regulator of the highly aggressive and frequently metastatic Group 3 medulloblastoma (MB). They show that Notch1+ cells represent the self-renewing population that gives rise to re-transplantable cerebellar tumors and primary spinal metastasis whereas Notch1- cells do not. Loss of function studies via shRNA-mediated knockdown and intrathecal Notch1 blocking antibody lead to lower frequency of spinal metastasis and better survival in mouse models while NICD1 expression rescues the phenotype. Finally, they provide evidence that Notch1 activates its downstream effector Bmi1 through activation of Twist1.

This study is relevant given that signaling mechanisms in grade 3 MBs are not well known and timely given the recent sequencing efforts on MB subtypes showing Notch pathway mutations in Group 3 MBs (Northcott et al, Nature 2017).

The manuscript is interesting and reports new data that will be of great interest to the field. The in vivo data are compelling and shows both loss, gain of function and rescue studies to ascertain Notch pathway function in group 3 MBs especially in metastasis and its potential value as a therapeutic target. While there is strong evidence for the role of Notch in MB metastasis, the “cancer stem cell” studies need to be supplemented, and it is not clear whether there is a distinction between the Notch1+ cancer stem cell subpopulation vs. the metastasis initiating population. Since the Notch signaling blockade appears to primarily inhibit spinal metastasis, it is possible that the metastatic initiators are a subclone of the Notch1+ tumor cells, consistent with the theory of multiple cancer stem cell clones, and it will be interesting to determine the genomic differences between these cells. The downstream signaling studies also need more biochemical experiments to show direct or indirect interactions.

The manuscript text as it is needs to be improved as it is difficult to read, with data connected to each other in different places (e.g. primary transplantation studies in Figure 1 then secondary transplants in Figure 3), and important details that were left out in the manuscript (e.g. no. of transplanted cells, how long were the mice treated with the Notch1 blocking antibody?). Some of the main figures also need to be broken up into separate figures so it conveys more specific points. Overall, however, this manuscript will be a great addition to the growing reports on MB molecular subtyping and to the field in general, although some changes and concerns need to be addressed.

Specific Comments:

1. In Figure 1:

- a. What is the status of other Notch1 downstream effectors other than Hes1 (Hey?) in primary vs. metastatic tumors?
- b. Are Notch1+ cells proliferative or quiescent? Notch1 staining in Figure 1g-primary is not

convincing. Does it express markers of other cancer/stem cells?

- c. 1a brain is upside down whereas rest of brain images are upright. Are there differences in rate of metastasis between human and mouse grade 3 MB primary tumor cells?
- d. No survival difference in Notch1^{high} vs low expressing cells---may be expected since Notch expression is low in primary tumors. Is there difference if all tumor samples are included (includes other subtypes) or if Notch1 expression is limited to metastatic tumors?
- e. Migration assay in Ext Figure 5 should be done using Notch1⁺ vs – cells as it is expected that metastatic cells are more migratory compared to primary tumor cells.
- f. What is the % of Notch 1⁺ vs. – cells in the primary transplanted tumors in Figure 1h-k?

2. In Figure 2:

- a. Does knockdown of Notch1 significantly affect cell viability, proliferation and differentiation of primary tumor cells (in vitro), or tumor grade, proliferative index, etc (in vivo)?
- b. How long was Notch1 blocking antibody treatment done (not clear based on the text) and what are side effects given that Notch1 is expressed in stem cell compartments and other tissues? Do transplanted MB cells fail to engraft or metastasize then undergo apoptosis upon inhibitor treatment?
- c. Is there data showing that the blocking antibody was able to reach its target and cause inhibition? Since the drug is administered intrathecally, it may not have been as efficiently delivered into the brain parenchyma compared to the spinal metastasis.
- d. Does NICD directly bind to the TWIST1 promoter and Twist1 to the BMI-1 promoter in Grade 3 MB cells? Chip-Seq should be able to clarify.
- e. What specific downstream targets of Bmi1 enables Notch1⁺ cells to metastasize?

3. In Figure 3:

- a. The first part of the serial transplantation experiment (primary) is a repeat of Figure 1h-k. These data(primary and secondary transplantations) are better presented in the same figure.
- b. How many Notch1⁺ vs – cells were transplanted in the primary tumor transplantation experiment? How many cells were transplanted in the secondary transplantation experiment? In vivo serial dilutions should provide threshold of tumorigenicity for + vs. – cells.
- c. Inhibition of Notch1 has a greater effect on spinal metastasis than cerebellar tumor growth, suggesting that a Notch-regulated subclone arises and leaves the brain parenchyma and seeds the spinal canal. Are there molecular differences between Notch1⁺ cerebellar vs. spinal tumor cells and do Notch1⁺ spinal tumors accumulate additional genetic changes? Shh medulloblastomas were previously shown to be genetically divergent between primary and metastatic tumors (Wu, Taylor, Nature 2012), but sequencing was done in bulk tumors, so it will be interesting to know differences in purer subpopulations.

Reviewer #3 (Remarks to the Author):

Group 3 medulloblastoma is the most aggressive of medulloblastoma, the most common pediatric brain tumor. Understanding the mechanisms driving metastatic disease in this tumor is critical.

In this elegant work, Kahn and colleagues demonstrate that the NOTCH1 signaling pathway regulates metastasis and self renewal/maintenance in these tumors, through NOTCH-induced BMI1 activation mediated by Twist1 activation. The experiments in this paper are well designed and convincing. The paper is also extremely well written and clear.

2 minor revisions:

1. In the human Group 3 medulloblastoma samples, can the authors show that NOTCH1 and BMI1 are overexpressed in patients that do develop metastases? Is there an increased frequency of metastases in patients with NOTCH1 and BMI1 overexpression or a decreased time to development of metastases?

2. Do the authors have any matched metastatic human samples showing NOTCH1 and BMI1 overexpression in the metastases compared to primary samples? Even a small number of cases would be helpful here.

Response to the Reviewers' Comments for Nature Communications manuscript NCOMMS-17-17499:

(Our responses in blue)

Reviewer #1 (Remarks to the Author):

This manuscript claims that Notch1, but not Notch2, is largely responsible for leptomeningeal spread of Group 3 medulloblastomas. The authors further claim that transcriptional induction of BMI1 through activation of Twist1 is the mechanism by which medulloblastoma cells gain metastatic potential and possibly self-renewal. The study is well done and some specific findings are novel even if the role of NOTCH in medulloblastoma has been debated for over a decade.

We thank the reviewer for acknowledging that our work is well done.

Several topics could be better addressed in this manuscript:

The use of models was inconsistent throughout the paper and in some cases problematic. No data is presented or cited to indicate that D283 or D425 cell lines are truly Group 3. Given that a frequently used medulloblastoma line (DAOY) is said to be isolated from a 4 year old boy, yet has two x chromosomes and no Y chromosomes, it is critical to validate these "workhorse" cell lines if they are used in papers intended to influence clinical trial design.

Thank you for your comment. We absolutely agree that cell lines need to be constantly validated, so we used NanoString (please see table below) to determine subgrouping and all medulloblastoma cell lines used in this study presented MYC amplification (Methods section, under "Medulloblastoma cell cultures").

Sample	D3S135 8	TH01	D21S11	D18S51	Penta E	D5S818	D13S31 7	D7S820	D16S53 9	CSF1PO	Penta D	vWA	D8S117 9	TPOX	FGA	AMEL
MB002	15, 16	6, 9	30, 32, 2	14, 14	5, 19	11, 12	8, 12	8, 12	10, 13	11, 12	9, 14	14, 14	12, 16	8, 8	23, 24	X, X
D425	14, 17	6, 9, 3	29, 33, 2	15, 18	14, 15	12, 13	11, 11	11, 12	11, 12	9, 12	9, 10	16, 17	11, 15	8, 8	20, 24	X, X
D283	15, 15	7, 7	28, 30, 2	16, 18	7, 10	11, 11	8, 10	10, 10	11, 11	9, 12	9, 13	16, 18, 19	12, 13	8, 11	20, 23	X, Y

There is no explanation why some cell lines or models, but not others, were used for certain experiments. If additional relevant data was generated from some of these models but not included in the manuscript, it should be added, even if inconsistent with the current narrative.

We thank the reviewer for this observation. This study started with the characterization of our in house patient-derived MB002 cell line, which presents low levels of spinal metastasis *in vivo*. We therefore started using the D425 and D283 cell lines, which are highly metastatic *in vivo*. To make sure that our findings were not cell line-specific, we confirmed most of our findings in D425 with other medulloblastoma lines (such as D283 and MB002), and these data are included in the manuscript. Due to space constraints, some of the results are shown as Extended Data.

The focus of this paper is on Group 3 medulloblastoma, but it is not clear whether there is a link to MYC/MYCN amplification, which is the subset of Group 3 tumors with the poorest prognosis. A link between the NOTCH pathway and MYC was previously made (PMID: 24708907).

Thank you for your helpful suggestion. We have now silenced *NOTCH1* and *TWIST1* in D425 cells and analyzed *MYC* expression. Please see figure below. *NOTCH1*-silenced medulloblastoma cells express lower levels of *MYC* (Extended Data Figure 2e). Interestingly, *TWIST1*-silenced medulloblastoma cells present similar levels of *MYC* expression as compared to control, suggesting that NOTCH1, but not TWIST1, is upstream of *MYC*. Although this finding indicates that NOTCH1-MYC and NOTCH1-TWIST1-BMI1 act at different axes in Group 3 medulloblastoma metastasis, further studies will be necessary to confirm this possibility.

The analysis of Group 3 medulloblastoma samples from the recently published Cavalli dataset (PMID: 28609654) also shows a direct correlation between *NOTCH1* and *MYC* expression. Please see figure below.

We also analyzed the expression of activated Notch1 and Hes1 in primary and metastatic tumors in a spontaneous MYCN-driven transgenic medulloblastoma mouse model (GTML) (PMID: 20478998), and found that spinal metastasis expresses higher levels of activated Notch1 and Hes1 compared to the primary tumor (Extended Data Fig. 4d).

Overexpression of ATOH1 caused SHH-driven medulloblastomas to form leptomeningeal disease and become more aggressive (PMID: 28490517). Given that this prior publication could diminish the novelty of the current manuscript, it seems important to either address whether activation of NOTCH1 in non-Group 3 medulloblastomas might lead to the less common leptomeningeal metastases cases in the other groups. While it is too much to ask for the current manuscript, it might be worth assessing NOTCH1 pathway in human leptomeningeal disease compared to primary tumor (autopsy cases) from all groups. An easier step for the current manuscript would be to assess notch pathway signatures from patients who were M+ vs M0 regardless of molecular subclass. The reviewer recognizes that the tissue derived from the primary tumor may not show the full extent of NOTCH pathway activation in mets.

Thank you, this is an excellent point. We have now quantified the expression of *NOTCH1* and the degree of expression variation across and within medulloblastoma subgroups in primary patient samples in the entire MAGIC medulloblastoma cohort. We found that *NOTCH1* is among the genes with the highest expression variation in medulloblastoma (88th percentile of all genes). Please see figure below.

Variation of expression of all genes, all medulloblastoma samples

Variation of Gene Expression

NOTCH1 is also significantly differentially expressed between subgroups of medulloblastoma, Group 3 vs Group 4 ($P = 0.025$) and Group 3 vs SHH ($P = 6.1 \times 10^{-3}$). Please see figure below.

Distribution of expression of NOTCH1 gene

We also analyzed *NOTCH1* expression in all Groups (Kool et al. 2008 dataset, a = WNT, b = SHH, c/d = Group 4, e = Group 3). Please see figure below.

Finally, we compared *NOTCH1* expression in Group 3 medulloblastoma patients, stratifying by presence (M+) or absence (M0) of metastasis at diagnosis (Cavalli Dataset PMID: 28609654) and found that *NOTCH1* expression is significantly higher in M+ patients. Please see figure below (Extended Data Figure 3d). This difference is not observed in other subgroups.

The authors argue that the failure of a pan-NOTCH inhibitor in human clinical trials was due to the lack of therapeutic window when both NOTCH1 and NOTCH2 are inhibited. This claim is refuted by a clinical trial in which NOTCH signal was diminished in 6 of 7 children for whom tissue was available 24 hours after MK-0752 dosing at doses that were well tolerated. Furthermore, bi-allelic knockout of NOTCH1 and NOTCH2 had no effect pathogenesis of Shh-driven medulloblastomas in murine models, suggesting that the pathway may be unimportant in some types of medulloblastoma (PMID: 20440271) and that clinical trials should focus specifically on molecularly selected patients who are most likely to benefit from NOTCH1 inhibition. With this in mind, it is important to add depth to the discussion with regard to the potential use of anti-NRR1 antibodies intrathecally in human patients with leptomeningeal disease. Intrathecal medications are not typically used in this population because CSF flow is impaired by metastases. Antibodies do not cross the BBB. So, what, if anything, are the authors proposing in terms of future clinical trials?

Thank you for your comment. We have now added depth to the discussion with regard to the potential use of anti-NRR1 in the clinic.

The effect that we observed of NOTCH1 inhibition in Group 3 medulloblastoma is on reduction of tumor metastasis. As the SHH mouse models are not metastatic and are molecularly distinct from Group 3 medulloblastoma, we do not expect to see a benefit from inhibiting NOTCH1 in these models.

As intrathecal methotrexate is the standard of care for infants in Europe (PMID: 15758008), and intraventricular administration maximizes the therapeutic dose in the cerebrospinal fluid, we propose intrathecal administration of anti-NRR1 as a delivery method in Group 3 medulloblastoma patients.

Likewise, for this work to be important in clinical translation, it would be important to know if anti-NRR1 antibodies affect metastases that have already formed. Prevention of metastases could be important, but effective treatment of established metastases would be much more valuable. Additional experiments showing the effect of anti-NRR1 on established metastases would be valuable if technically feasible given the impaired CSF flow and the challenges of quantifying leptomeningeal disease burden before and after treatment in living mice. Perhaps inducible knock-out/knock-down would be a feasible method.

We thank the reviewer for this excellent observation, which motivated us to administer anti-NRR1 intrathecally in mice with established Group 3 medulloblastoma metastasis. Although the effect of anti-NRR1 is less evident when mice are treated after spinal metastases are formed, spinal metastases-bearing mice intrathecally treated with anti-NRR1 presented lower frequency of spinal metastases and higher survival rate. Please see figure below (Extended Data Fig. 7g, h).

The statement that elevated NOTCH1 and HES1 “trended” toward worse survival in patients involved in the MAGIC consortium lacks statistical and biological rigor. Was this observed in other large datasets? Why was HES1 chosen as the only NOTCH1 target gene for this analysis? Given that HES1 is activated by other forms of NOTCH and that there are many other targets of NOTCH1, the selection of HES1 seems somewhat arbitrary. Since the intracellular domain of NOTCH is the active domain, what is the link between gene expression and NICD protein fragment?

Indeed, there is no significant difference between the survival curves of *NOTCH1* high versus low medulloblastoma patients. Similarly, there is no significant difference between the survival curves of *HEY1* (another gene known to be activated by NOTCH/RBP-J signaling) high versus low medulloblastoma patients. Please see below the analysis of the Cavalli dataset (PMID: 28609654).

We would like to mention that normal human cerebella express high levels of *NOTCH1* (please see figure below), and the surgical samples used to create these expression databases were not processed to separate cancer cells from non-cancer cells within the tumor bulk. Thus, any bulk tumor analysis will be confounded by contaminating non-tumor cells, probably contributing to the lack of statistically significant levels of expression in these analyses.

“N cerebellum” = normal cerebellum.

Regarding the correlation between NOTCH1 activation and gene expression, we have performed comparative analysis of NOTCH1 pathway genes expression in:

- Medulloblastoma cells sorted from primary tumors: shNotch1-Dox versus shNotch1+Dox (Extended Data Table 2).
- Medulloblastoma cells sorted from primary tumors: anti-NRR1 versus control (Extended Data Table 3).

In summary, interesting, important and experimentally well supported conclusions with potential to influence how clinical trialists and biologists view medulloblastoma leptomeningeal disease.

Reviewer #2 (Remarks to the Author):

Kahn et al. describe the role of Notch signaling as a regulator of the highly aggressive and frequently metastatic Group 3 medulloblastoma (MB). They show that Notch1+ cells represent the self-renewing population that gives rise to re-transplantable cerebellar tumors and primary spinal metastasis whereas Notch1- cells do not. Loss of function studies via shRNA-mediated knockdown and intrathecal Notch1 blocking antibody lead to lower frequency of spinal metastasis and better survival in mouse models while NICD1 expression rescues the phenotype. Finally, they provide evidence that Notch1 activates its downstream effector Bmi1 through activation of Twist1.

This study is relevant given that signaling mechanisms in grade 3 MBs are not well known and timely given the recent sequencing efforts on MB subtypes showing Notch pathway mutations in Group 3 MBs (Northcott et al, Nature 2017).

We thank the reviewer for acknowledging that our work is relevant and timely.

The manuscript is interesting and reports new data that will be of great interest to the field. The in vivo data are compelling and shows both loss, gain of function and rescue studies to ascertain Notch pathway function in group 3 MBs especially in metastasis and its potential value as a therapeutic target. While there is strong evidence for the role of Notch in MB metastasis, the “cancer stem cell” studies need to be supplemented, and it is not clear whether there is a distinction between the Notch1+ cancer stem cell subpopulation vs. the metastasis initiating population. Since the Notch signaling blockade appears to primarily inhibit spinal metastasis, it is possible that the metastatic initiators are a subclone of the Notch1+ tumor cells, consistent with the theory of multiple cancer stem cell clones, and it will be interesting to determine the genomic differences between these cells. The downstream signaling studies also need more biochemical experiments to show direct or indirect interactions.

Thank you for your excellent points. We address all your observations in “Specific Comments” below.

The manuscript text as it is needs to be improved as it is difficult to read, with data connected to each other in different places (e.g. primary transplantation studies in Figure 1 then secondary transplants in Figure 3), and important details that were left out in the manuscript (e.g. no. of transplanted cells, how long were the mice treated with the Notch1 blocking antibody?). Some of the main figures also need to be broken up into separate figures so it conveys more specific points. Overall, however, this manuscript will be a great addition to the growing reports on MB molecular subtyping and to the field in general, although some changes and concerns need to be addressed.

We thank the referee for these helpful suggestions. We have now:

- specified the number of medulloblastoma cells injected into mice cerebella in the Methods section, under “Orthotopic transplantation of medulloblastoma cells”;
- specified the anti-NRR1 treatment period in the Methods section, under “Intrathecal treatment”.

We experimented transferring the primary transplantation data to Figure 3, however the resulting organization negatively affected the flow of the manuscript. So we decided to keep the primary transplantation data in Figure 1.

Specific Comments:

1. In Figure 1:

a. What is the status of other Notch1 downstream effectors other than Hes1 (Hey?) in primary vs. metastatic tumors?

We have performed comparative analysis of NOTCH pathway genes expression in medulloblastoma cells sorted from primary and metastatic tumors. The complete list of genes over-expressed and under-expressed in spinal metastasis as compared to primary tumor from xenograft is in the Extended Data Table 1. *HEY1* and *HEY2*, for example, are over-expressed in spinal metastasis.

b. Are Notch1+ cells proliferative or quiescent? Notch1 staining in Figure 1g-primary is not convincing. Does it express markers of other cancer/stem cells?

Notch1+ cells are proliferative (please see figure below). For real-time analysis, IncuCyte Zoom System (Essen Biosciences, Ann Arbor, MI, USA) was utilized.

We thank the referee for this comment, which motivated us to analyze NOTCH1 expression in primary Group 3 medulloblastoma biopsy samples from five patients. Normal cerebellum was used as positive control. Please see Figure below (Extended Data Figure 3b). NOTCH1 is weakly expressed in all human primary tumors in the cerebellum, confirming the low the percentage of NOTCH1+ cells in the primary tumor site (Figures 1c-g and Extended Data Figure 1).

Regarding the expression of other stem/progenitor cell markers, we have now analyzed BMI1 and Nestin expression in primary tumors from Group 3 medulloblastoma-bearing mice. Please see figure below (Extended Data Figure 12).

Also, regarding cancer stem cell properties, NOTCH1+ cells present higher neurosphere forming ability (Extended Data Figure 11) and self-renewing capacity (Figure 3c) than NOTCH1- cells.

c. 1a brain is upside down whereas rest of brain images are upright. Are there differences in rate of metastasis between human and mouse grade 3 MB primary tumor cells?

Thank you, Figure 1a has been flipped.

In humans, 40-50% Group 3 medulloblastoma are metastatic. In our MB002 mouse xenograft model, approximately 40% of the mice present metastasis. In D425 and D283 models, approximately 90-100% of the mice present metastasis. Mouse Group 3 MB primary tumor cells from MP-1 somatic mouse model shows 80% metastatic penetrance whereas the GTML spontaneous model shows only 13-15% metastatic penetrance.

d. No survival difference in Notch1high vs low expressing cells---may be expected since Notch expression is low in primary tumors. Is there difference if all tumor samples are included (includes other subtypes) or if Notch1 expression is limited to metastatic tumors?

Indeed, there is no significant difference between the survival curves of *NOTCH1* high versus low medulloblastoma patients. Similarly, there is no significant difference between the survival curves of *NOTCH1* high versus low if all subtypes are included nor if *NOTCH1* expression is limited to metastatic tumors. Please see figures below.

We would like to mention that normal human cerebella express high levels of NOTCH1 (please see figure below), and the surgical samples used to create these expression databases were not processed to separate cancer cells from non-cancer cells within the tumor bulk. Thus, any bulk tumor analysis will be confounded by contaminating non-tumor cells, probably contributing to the lack of statistically significant levels of expression in these analyses.

"N cerebellum" = normal cerebellum.

e. Migration assay in Ext Figure 5 should be done using Notch1+ vs – cells as it is expected that metastatic cells are more migratory compared to primary tumor cells.

Thank you for your suggestion. We have now performed matrigel invasion assays with NOTCH1- versus NOTCH1+ medulloblastoma cells (Extended Data Figure 5d).

f. What is the % of Notch 1+ vs. – cells in the primary transplanted tumors in Figure 1h-k.

We thank the reviewer for this excellent question. After dissociation of the primary tumors generated by NOTCH1+ and NOTCH1- medulloblastoma cells, tumor cells were resorted based on surface NOTCH1 expression and re injected into mouse cerebella (second *in vivo* passage). Before the second passage injections, we analyzed the % of NOTCH1+ versus NOTCH1- cells in these tumors. Interestingly, mice originally injected with NOTCH1- cells generated tumors with 96.9% NOTCH1- cells, while mice originally injected with NOTCH1+ cells generated tumors with 81.2% NOTCH1+ cells (almost 20% NOTCH1- cells), indicating a cell lineage of self-renewing NOTCH1+ cells capable of giving rise to both NOTCH1+ and NOTCH1- cells, whereas NOTCH1- cells can only give rise to other NOTCH1- cells (Extended Data Figure 13).

To avoid confusion, we substituted “First passage” to “Before first passage injections” and “Second Passage” to “Before second passage injections” in Extended Data Figure 13.

2. In Figure 2:

a. Does knockdown of Notch1 significantly affect cell viability, proliferation and differentiation of primary tumor cells (in vitro), or tumor grade, proliferative index, etc (in vivo)?

We have analyzed the effect of NOTCH1 knockdown in medulloblastoma cells viability (Extended Data Figure 10). We compared the effects of NOTCH1 knockdown to NOTCH2 knockdown. NOTCH1-silenced medulloblastoma cells show some early apoptotic cells (Annexin V+ / DAPI-) cells whereas NOTCH2-silenced medulloblastoma cells present a higher percentage of late apoptotic cells (Annexin V+ / DAPI+) cells. WST-1 cell viability analysis of NOTCH1-silenced and NOTCH2-silenced D425 and MB002 cells showed that NOTCH1-silenced cells presented lower viability as compared to control cells, and that NOTCH2-silenced medulloblastoma cells presented lower viability than NOTCH1-silenced cells. Of note, NOTCH1-silenced medulloblastoma cells remain viable through multiple passages without regaining NOTCH1 expression.

b. How long was Notch1 blocking antibody treatment done (not clear based on the text) and what are side effects given that Notch1 is expressed in stem cell compartments and other tissues? Do transplanted MB cells fail to engraft or metastasize then undergo apoptosis upon inhibitor treatment?

Thank you for your question. Once tumor masses were detected in the brain, mice were randomized in two groups based on flux values, prior to osmotic pumps implantation. The pumps containing anti-NRR1 stayed subcutaneously on the dorsum, with intraventricular access, until mice were sacrificed (between 10 and 50

days). We have now added more detailed information about anti-NRR1 treatment timing in the Methods section, under “Intrathecal treatment”. We did not notice treatment-related sequelae in anti-NRR1-treated mice.

Anti-NRR1 treatment started after confirmation of tumor engraftment. After primary tumors were detected by bioluminescence imaging, mice were randomized in two groups and one group was treated with anti-NRR1. Mice treated with anti-NRR1 presented lower frequency of metastasis (Fig. 2e-g and Extended Data Fig. 7d,e), and showed similar levels of cell death (Please see figure below), as compared to control. Also, as shown in Extended Data Figure 10, NOTCH1-silenced medulloblastoma cells are slightly less viable as compared to controls.

c. Is there data showing that the blocking antibody was able to reach its target and cause inhibition? Since the drug is administered intrathecally, it may not have been as efficiently delivered into the brain parenchyma compared to the spinal metastasis.

Tumor growth in this model occurs mostly in the leptomeningeal CSF spaces, same location as the anti-NRR1 antibody was injected. Intrathecal administration maximizes the amount of antibody in the ventricles and central canal, where most of the primary and metastatic tumors are located, respectively. Moreover, we have previously shown, in the same metastatic medulloblastoma models (MB002, D425 and D283), that tumor-targeting monoclonal antibodies (PMID: 28298418) reach the brain parenchyma.

d. Does NICD directly bind to the TWIST1 promoter and Twist1 to the BMI-1 promoter in Grade 3 MB cells? Chip-Seq should be able to clarify.

We thank the reviewer for this comment. By protein expression and *in vivo* analysis, we show the functional involvement of TWIST1 and BMI1 in NOTCH1-induced medulloblastoma metastasis. Our data show that NOTCH1-induced medulloblastoma metastasis occurs through the activation of the NICD1-TWIST1-BMI1 signaling axis (Fig. 3i-k). Whether this pathway has intermediate players, or NICD1 directly binds to TWIST1 promoter and TWIST1 to BMI1 promoter is still unknown, and is not the focus of the current manuscript. However, this is an interesting avenue to explore in a follow up work. We have now added a discussion on this subject in the manuscript.

e. What specific downstream targets of Bmi1 enables Notch1+ cells to metastasize?

Glinksy et al (PMID: 15931389) reported that BMI1-associated gene expression pathway is activated in metastatic tumors. The authors show that expression of an 11-gene signature is a powerful predictor of a short

interval to distant metastasis and poor survival after therapy in breast and lung cancer patients diagnosed with an early-stage disease. CCNB1 (PMID 18048386), BUB1 (PMID: 18442402, 26287798), USP22 (PMID: 28427243, 27145278), RNF2 (PMID: 26450788, 27911266), FGFR2 (PMID: 17145872), which are known to be involved in tumor metastasis are in the 11-gene signature.

3. In Figure 3:

a. The first part of the serial transplantation experiment (primary) is a repeat of Figure 1h-k. These data (primary and secondary transplantations) are better presented in the same figure.

We experimented transferring the primary transplantation data to Figure 3, however the resulting organization negatively affected the flow of the manuscript. So we decided to keep the primary transplantation data in Figure 1.

b. How many Notch1+ vs – cells were transplanted in the primary tumor transplantation experiment? How many cells were transplanted in the secondary transplantation experiment? In vivo serial dilutions should provide threshold of tumorigenicity for + vs. – cells.

In the primary and secondary transplantation experiments, 30,000 medulloblastoma cells were injected into mice cerebella. We did perform an experiment using a more limited number of cells in Figure 3g-l (100 medulloblastoma cells per mouse), where the objective was to assess the tumorigenicity of a more limited number of medulloblastoma cells, demonstrating that NOTCH1+ cells have greater tumor forming ability *in vivo* than NOTCH1- cells.

c. Inhibition of Notch1 has a greater effect on spinal metastasis than cerebellar tumor growth, suggesting that a Notch-regulated subclone arises and leaves the brain parenchyma and seeds the spinal canal. Are there molecular differences between Notch1+ cerebellar vs. spinal tumor cells and do Notch1+ spinal tumors accumulate additional genetic changes? Shh medulloblastomas were previously shown to be genetically divergent between primary and metastatic tumors (Wu, Taylor, Nature 2012), but sequencing was done in bulk tumors, so it will be interesting to know differences in purer subpopulations.

This is a very interesting point. We would not expect to observe accumulation of additional genetic changes due to the brief time window in which metastases occur in animal models. In accordance with previous studies (PMID: 25689980), analysis of human Group 3 medulloblastoma samples demonstrated similar methylation profiles in primary and metastatic tumor bulks (Please see figure below). We analyzed normalized 450k DNA methylation (Infinium HumanMethylation450 BeadChip Kit) data from 27 medulloblastoma samples (11 matched primary-metastasis patients) using 55 methylated probes attributed to the gene *NOTCH1*. We were not able to identify any distinct molecular differences between the primary and metastatic compartments.

We would like to mention that the surgical samples used to create these databases were not processed to separate cancer cells from non-cancer cells within the tumor bulk. Thus, any bulk tumor analysis will be confounded by contaminating non-tumor cells.

Reviewer #3 (Remarks to the Author):

Group 3 medulloblastoma is the most aggressive of medulloblastoma, the most common pediatric brain tumor. Understanding the mechanisms driving metastatic disease in this tumor is critical.

In this elegant work, Kahn and colleagues demonstrate that the NOTCH1 signaling pathway regulates metastasis and self renewal/maintenance in these tumors, through NOTCH-induced BMI1 activation mediated by Twist1 activation. The experiments in this paper are well designed and convincing. The paper is also extremely well written and clear.

We thank the reviewer for the positive remarks on our study as a whole.

2 minor revisions:

1. In the human Group 3 medulloblastoma samples, can the authors show that NOTCH1 and BMI1 are overexpressed in patients that do develop metastases? Is there an increased frequency of metastases in patients with NOTCH1 and BMI1 overexpression or a decreased time to development of metastases?

Thank you for your suggestion. We compared *NOTCH1* expression in Group 3 medulloblastoma patients stratifying by presence (M+) or absence (M0) of metastasis at diagnosis (Cavalli dataset), and found that *NOTCH1* expression is higher in M+ patients. Please see figure below (Extended Data Figure 3d). BMI1, on the other hand, is not significantly different between M0 and M+ patients.

We would like to mention that normal human cerebella express high levels of NOTCH1 (please see figure below), and the surgical samples used to create these expression databases were not processed to separate cancer cells from non-cancer cells within the tumor bulk. Thus, any bulk tumor analysis will be confounded by contaminating non-tumor cells, probably contributing to the lack of statistically significant levels of expression in these analyses.

“N cerebellum” = normal cerebellum.

2. Do the authors have any matched metastatic human samples showing NOTCH1 and BMI1 overexpression in the metastases compared to primary samples? Even a small number of cases would be helpful here.

Yes, spinal metastasis samples show higher NOTCH1 expression compared to primary tumors from the same patients (Extended Data Figure 3a).

Accordingly, NOTCH1 is weakly expressed in primary Group 3 medulloblastoma biopsy samples from five patients. Please see Figure below (Extended Data Figure 3b). Normal cerebellum was used as positive control.

Reviewers' comments:

Reviewer #1 (Remarks to the Author):

This reviewer feels that the investigators have adequately addressed the concerns of the prior reviews.

Reviewer #2 (Remarks to the Author):

Kahn et al. addressed most of the concerns that were put forward and substantially added to the data and clarified details, which enhanced the manuscript.

However, there are a few issues that remain unresolved (original requests and authors response in blue; new reviewer comments in red (EDITORS NOTE- NEW COMMENTS IN ITALICS)):

1. Figure 1, comment b:

Are Notch1+ cells proliferative or quiescent? Notch1 staining in Figure 1g-primary is not convincing. Does it express markers of other cancer/stem cells?

We thank the referee for this comment, which motivated us to analyze NOTCH1 expression in primary Group 3 medulloblastoma biopsy samples from five patients. Normal cerebellum was used as positive control.

Please see Figure below (Extended Data Figure 3b). NOTCH1 is weakly expressed in all human primary tumors in the cerebellum, confirming the low percentage of NOTCH1+ cells in the primary tumor site (Figures 1c-g and Extended Data Figure 1).

Regarding the expression of other stem/progenitor cell markers, we have now analyzed BMI1 and Nestin expression in primary tumors from Group 3 medulloblastoma-bearing mice. Please see figure below (Extended Data Figure 12).

 The question was with regards to the stem cell properties of Notch1+ cells specifically--- whether they are proliferative or quiescent in vivo (e.g. by co-labeling of Notch1 with Ki67/other proliferation markers, performing BrdU pulse chase in mouse models), and not just in vitro, as it is expected that these cells proliferate in culture. We are also looking for the co-expression of Notch1+ cells with markers of other cancer/stem cells (by immunofluorescence staining, FACS), not just in whole tumor tissue. As it stands, the authors' response is indirect and correlative. 

2. In Figure 3c:

Inhibition of Notch1 has a greater effect on spinal metastasis than cerebellar tumor growth, suggesting that a Notch-regulated subclone arises and leaves the brain parenchyma and seeds the spinal canal. Are there molecular differences between Notch1+ cerebellar vs. spinal tumor cells and do Notch1+ spinal tumors accumulate additional genetic changes? Shh medulloblastomas were previously shown to be genetically divergent between primary and metastatic tumors (Wu, Taylor, Nature 2012), but sequencing was done in bulk tumors, so it will be interesting to know differences in purer subpopulations.

This is a very interesting point. We would not expect to observe accumulation of additional genetic changes due to the brief time window in which metastases occur in animal models. In accordance with previous studies (PMID: 25689980), analysis of human Group 3 medulloblastoma samples demonstrated similar methylation profiles in primary and metastatic tumor bulks (Please see figure

below). We analyzed normalized 450k DNA methylation (Infinium HumanMethylation450 BeadChip Kit) data from 27 medulloblastoma samples (11 matched primary-metastasis patients) using 55 methylated probes attributed to the gene NOTCH1. We were not able to identify any distinct molecular differences between the primary and metastatic compartments. We would like to mention that the surgical samples used to create these databases were not processed to separate cancer cells from non-cancer cells within the tumor bulk. Thus, any bulk tumor analysis will be confounded by contaminating non-tumor cells.

As mentioned, the bulk tumor analysis may not show any difference because of the heterogeneity of the tumor sample preparation. The system the authors use allow them to separate the Notch1+/- cells from primary vs. spinal metastasis, and in light of divergent results from different studies, should be analyzed for genomic expression and mutational differences. This is also in connection with the issue raised in the general comments, where we noted that it was not clear what the distinction was between the Notch1+ cancer stem cell subpopulation vs. the metastasis initiating population. The authors show increased Notch pathway activation in metastatic cells as compared to primary cells, and an increased % of Notch1+ cells, going from a small subset of the primary tumor to a majority of cells in the metastatic clones. Hence, whether these two populations (Notch1+ primary cells vs. Notch1+ metastatic cells) contain the same genomic changes or evolved to become more infiltrative, or represent the same population of cells need to be addressed.

Overall, the manuscript the authors show compelling in vivo data that will present an advancement to the medulloblastoma literature in particular and the cancer field in general.

Reviewer #3 (Remarks to the Author):

The authors have adequately addressed my comments and significantly strengthened the paper.

Manuscript tracking number: NCOMMS-17-17499B.

Response to reviewer's comments in green

Reviewers' comments:

Reviewer #1 (Remarks to the Author):

This reviewer feels that the investigators have adequately addressed the concerns of the prior reviews.

Thank you.

Reviewer #2 (Remarks to the Author):

Kahn et al. addressed most of the concerns that were put forward and substantially added to the data and clarified details, which enhanced the manuscript.

However, there a few issues that remain unresolved (original requests and authors response in blue; new reviewer comments in red):

1. Figure 1, comment b:

Are Notch1+ cells proliferative or quiescent? Notch1 staining in Figure 1g-primary is not convincing. Does it express markers of other cancer/stem cells?

Notch1+ cells are proliferative (please see figure below). For real-time analysis, IncuCyte Zoom System (Essen Biosciences, Ann Arbor, MI, USA) was utilized.

We thank the referee for this comment, which motivated us to analyze NOTCH1 expression in primary Group 3 medulloblastoma biopsy samples from five patients. Normal cerebellum was used as positive control.

Please see Figure below (Extended Data Figure 3b). NOTCH1 is weakly expressed in all human primary tumors in the cerebellum, confirming the low the percentage of NOTCH1+ cells in the primary tumor site (Figures 1c-g and Extended Data Figure 1).

Regarding the expression of other stem/progenitor cell markers, we have now analyzed BMI1 and Nestin expression in primary tumors from Group 3 medulloblastoma-bearing mice. Please see figure below (Extended Data Figure 12).

Also, regarding cancer stem cell properties, NOTCH1+ cells present higher neurosphere forming ability (Extended Data Figure 11) and self-renewing capacity (Figure 3c) than NOTCH1- cells.

The question was with regards to the stem cell properties of Notch1+ cells specifically--- whether they are proliferative or quiescent in vivo (e.g. by co-labeling of Notch1 with Ki67/other proliferation markers, performing BrdU pulse chase in mouse models), and not just in vitro, as it is expected that these cells proliferate in culture. We are also looking for the co-expression of Notch1+ cells with markers of other cancer/stem cells (by immunofluorescence staining, FACS), not just in whole tumor tissue. As it stands, the authors' response is indirect and correlative.

Thank you for the clarification. We have now co-stained primary medulloblastoma tumors with NOTCH1 / Ki67 and with NOTCH1 / CD15. D425 medulloblastoma cells were engineered for constitutive expression of GFP and luciferase and orthotopically injected into the cerebella of NSG mice. After confirmation of tumor growth by bioluminescent imaging, we determined the relative expression of Ki67 and CD15 on NOTCH1+ as compared to NOTCH1- medulloblastoma cells. We found that NOTCH1+ medulloblastoma cells from primary tumors express higher levels of CD15 (Extended Data Fig. 11c) and Ki67 (Extended Data Fig. 11d) than NOTCH1- medulloblastoma cells from the same tumors. Please see images below.

2. In Figure 3c:

Inhibition of Notch1 has a greater effect on spinal metastasis than cerebellar tumor growth, suggesting that a Notch-regulated subclone arises and leaves the brain parenchyma and seeds the spinal canal. Are there molecular differences between Notch1+ cerebellar vs. spinal tumor cells and do Notch1+ spinal tumors accumulate additional genetic changes? Shh medulloblastomas were previously shown to be genetically divergent between primary and metastatic tumors (Wu, Taylor, Nature 2012), but sequencing was done in bulk tumors, so it will be interesting to know differences in purer subpopulations.

This is a very interesting point. We would not expect to observe accumulation of additional genetic changes due to the brief time window in which metastases occur in animal models. In accordance with previous studies (PMID: 25689980), analysis of human Group 3 medulloblastoma samples demonstrated similar methylation profiles in primary and metastatic tumor bulks (Please see figure below). We analyzed normalized 450k DNA methylation (Infinium HumanMethylation450 BeadChip Kit) data from 27 medulloblastoma samples (11 matched primary-metastasis patients) using 55 methylated probes attributed to the gene NOTCH1. We were not able to identify any distinct molecular differences between the primary and metastatic compartments.

We would like to mention that the surgical samples used to create these databases were not processed to separate cancer cells from non-cancer cells within the tumor bulk. Thus, any bulk tumor analysis will be confounded by contaminating non-tumor cells.

As mentioned, the bulk tumor analysis may not show any difference because of the heterogeneity of the tumor sample preparation. The system the authors use allow them to separate the Notch1+/- cells from primary vs. spinal metastasis, and in light of divergent results from different studies, should be analyzed for genomic expression and mutational differences. This is also in connection with the issue raised in the general comments, where we noted that it was not clear what the distinction was between the Notch1+ cancer stem cell subpopulation vs. the metastasis initiating population. The authors show increased Notch pathway activation in metastatic cells as compared to primary cells, and an increased % of Notch1+ cells, going from a small subset of the primary tumor to a majority of cells in the metastatic clones. Hence, whether these two populations (Notch1+ primary cells vs. Notch1+ metastatic cells) contain the same genomic changes or evolved to become more infiltrative, or represent the same population of cells need to be addressed.

Overall, the manuscript the authors show compelling in vivo data that will present an advancement to the medulloblastoma literature in particular and the cancer field in general.

Thank you. We would not expect to observe accumulation of additional genetic changes due to the brief time window in which metastases occur in animal models. Unfortunately, because the percentage of NOTCH1+ medulloblastoma cells at the primary tumor site is very low, we were unsuccessful at sorting enough viable cells for genomic analysis. In our last try, we combined NOTCH1+ cells isolated from three different mice (3 consecutive shifts of 10 hour-sorting each), however we were again unsuccessful at obtaining enough material for genomic analysis. Therefore, whether NOTCH1+ primary and NOTCH1+ metastatic medulloblastoma cells represent the same population of cells, or contain genomic differences is still unknown and is an interesting avenue to explore in a follow up work. We have now added a discussion on this subject in the manuscript.

Reviewer #3 (Remarks to the Author):

The authors have adequately addressed my comments and significantly strengthened the paper.

Thank you.

REVIEWERS' COMMENTS:

Reviewer #1 (Remarks to the Author):

I feel that, within reasonable expectations, the authors have addressed the concerns of all three reviewers. The single set of experiments that were not completed in response to Reviewer #2 were attempted with persistence. Unfortunately, the proposed experiments do not appear to be technically feasible.